# Decoupled Training for Long-Tailed Classification With Stochastic Representations

**Giung Nam**[1*] **Sunguk Jang**[2*†] **Juho Lee**[1,2]

[1]Korea Advanced Institute of Science and Technology (KAIST), [2]AITRICS
[1]{giung, juholee}@kaist.ac.kr

## Abstract

Decoupling representation learning and classifier learning has been shown to be effective in classification with long-tailed data. There are two main ingredients in constructing a decoupled learning scheme; 1) how to train the feature extractor for representation learning so that it provides generalizable representations and 2) how to re-train the classifier that constructs proper decision boundaries by handling class imbalances in long-tailed data. In this work, we first apply Stochastic Weight Averaging (SWA), an optimization technique for improving the generalization of deep neural networks, to obtain better generalizing feature extractors for long-tailed classification. We then propose a novel classifier re-training algorithm based on stochastic representation obtained from the SWA-Gaussian, a Gaussian perturbed SWA, and a self-distillation strategy that can harness the diverse stochastic representations based on uncertainty estimates to build more robust classifiers. Extensive experiments on CIFAR10/100-LT, ImageNet-LT, and iNaturalist-2018 benchmarks show that our proposed method improves upon previous methods both in terms of prediction accuracy and uncertainty estimation.

## 1 Introduction

While deep neural networks have achieved remarkable performance on various computer vision benchmarks (e.g., image classification (Russakovsky et al., 2015) and object detection (Lin et al., 2014)), there still are many challenges when it comes to applying them for real-world applications. One of such challenges is that the real-world classification data are *long-tailed* - the distribution of class frequencies exhibits a long tail, and many of the classes have only a few observations belonging to them. As a consequence, the class distribution of such data is extremely imbalanced, degrading the performance of a standard classification model trained with the balanced class assumption due to a paucity of samples from tail classes (Van Horn et al., 2018; Liu et al., 2019). Thus, it is worth exploring a novel technique dealing with long-tailed data for real-world deployments. While several works have diagnosed the performance bottleneck of long-tailed recognition as distinct from balanced one (e.g., improper decision boundaries over the representation space (Kang et al., 2020), low-quality representations from the feature extractor (Samuel and Chechik, 2021)), the shared design principle of them is *giving tail classes a chance to compete with head classes*.

Decoupling (Kang et al., 2020) is one of the learning strategies proven to be effective for long-tailed data, where the representation learning via the feature extractor and classifier learning via the last classification layer are decoupled. Even for a classification network failing on long-tailed data, the representations obtained from the penultimate layer can be flexible and generalizable, provided that the feature extractor part is expressive enough (Donahue et al., 2014; Zeiler and Fergus, 2014; Girshick et al., 2014). The main motivation behind the *decoupling* is that the performance bottleneck of the long-tailed classification is due to the improper decision boundaries set over the representation space. Based on this, Kang et al. (2020) has shown that a simple re-training of the last layer parameters could significantly improve the performance.

---

[*] Equal contribution
[†] The work was done while the author was a graduate student at KAIST.

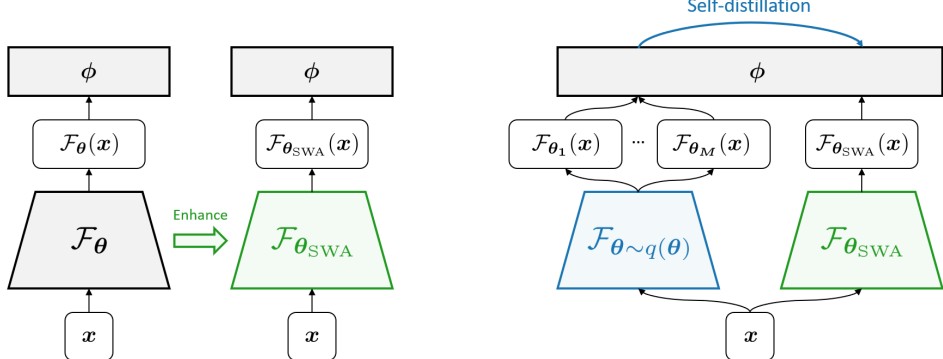

**Figure 1:** A schematic diagram depicting the overall composition of the paper. **Left:** We first apply SWA to obtain more generalizing feature extractor (Section 3). **Right:** We then propose a novel self-distillation strategy distilling SWAG into SWA to obtain more robust classifier (Section 4).

The success of decoupling naturally motivates obtaining more informative representations from which the classifier re-training can benefit. To this end, we first investigate Stochastic Weight Averaging (SWA; Izmailov et al., 2018), which improves the generalization performance of deep neural networks by seeking flat minima in loss surfaces. Although SWA has been successful for various tasks involving deep neural networks (for instance, supervised learning (Izmailov et al., 2018), semi-supervised learning (Athiwaratkun et al., 2019) and domain generalization (Cha et al., 2021)), to the best of our knowledge, it has never been explored for long-tailed classification problems. In Section 3, we empirically show that a naïve application of SWA for long-tailed classification would fail due to a similar bottleneck issue, but when combined with decoupling, SWA significantly improves the classification performance due to its property to obtain generalizable representations.

Confirming that SWA can benefit long-tailed classification, we take a step further and propose a novel classifier re-training strategy. For this, we first obtain *stochastic representations*, the output of penultimate layers computed with multiple feature extractor parameters drawn from an approximate posterior, where we construct the approximate parameter with SWA-Gaussian (SWAG; Maddox et al., 2019); SWAG is a Bayesian extension of SWA, adding Gaussian noise to the parameter obtained from SWA to approximate posterior parameter uncertainty. In Section 3.2, we first show that the diverse stochastic representations obtained from SWAG samples well reflect the uncertainty of inputs. Hinging on this observation, we propose a novel self-distillation algorithm where the stochastic representations are used to construct an ensemble of *virtual* teachers, and the classifier re-training is formulated as a distillation (Hinton et al., 2015) from the virtual teacher.

Fig. 1 depicts the overall composition of this paper as a diagram. Using CIFAR10/100-LT (Cao et al., 2019), ImageNet-LT (Liu et al., 2019), and iNaturalist-2018 (Van Horn et al., 2018) benchmarks for long-tailed image classification, we empirically validate that our proposed method improves upon previous approaches both in terms of prediction accuracy and uncertainty estimation.

## 2 PRELIMINARIES

### 2.1 DECOUPLED LEARNING FOR LONG-TAILED CLASSIFICATION

Let $\mathcal{F}_{\boldsymbol{\theta}} : \mathbb{R}^D \to \mathbb{R}^L$ be a neural network parameterized by $\boldsymbol{\theta}$ that produces $L$-dimensional outputs for given $D$-dimensional inputs. For the $K$-way classification problem, an output from $\mathcal{F}_{\boldsymbol{\theta}}$ is first transformed into $K$-dimensional logits via a linear classification layer parameterized by $\boldsymbol{\phi} = (\boldsymbol{w}_k \in \mathbb{R}^L, b_k \in \mathbb{R})_{k=1}^K$, and then turned into a classification probability with the softmax function,

$$\boldsymbol{p}^{(k)}(\boldsymbol{x}; \boldsymbol{\Theta}) = \frac{\exp\left(\boldsymbol{w}_k^\top \mathcal{F}_{\boldsymbol{\theta}}(\boldsymbol{x}) + b_k\right)}{\sum_{j=1}^K \exp\left(\boldsymbol{w}_j^\top \mathcal{F}_{\boldsymbol{\theta}}(\boldsymbol{x}) + b_j\right)}, \quad \text{for } k = 1, ..., K, \tag{1}$$

where $\boldsymbol{\Theta} = (\boldsymbol{\theta}, \boldsymbol{\phi})$ denotes a set of trainable parameters. Given a training set $\mathcal{D}$ consisting of pairs of input $\boldsymbol{x} \in \mathbb{R}^D$ and corresponding label $y \in \{1, \ldots, K\}$, $\boldsymbol{\Theta}$ is trained to minimize the cross-entropy

loss over $\mathcal{D}$,

$$\Theta^* = (\theta^*, \phi^*) = \arg\min_{\Theta} \mathbb{E}_{(\boldsymbol{x}, y) \sim \mathcal{D}} \left[ -\log \boldsymbol{p}^{(y)}(\boldsymbol{x}; \Theta) \right]. \tag{2}$$

Throughout the paper, we call $\mathcal{F}_{\boldsymbol{\theta}}$ with parameters $\boldsymbol{\theta}$ as a *feature extractor*, and the last linear layer with parameters $\phi$ as *classifier*. Also, we refer to a basic algorithm training the feature extractor and classifier together with Stochastic Gradient Descent (SGD; Robbins and Monro, 1951) as SGD.

Previous works (Liu et al., 2019; Van Horn et al., 2018) have shown that the vanilla SGD suffers from the data paucity of tail classes when applied to long-tailed classification tasks. Notably, Kang et al. (2020) showed that a simple re-training procedure on the classifier effectively resolves this problem. For instance, using pre-trained $\boldsymbol{\theta}^*$ from Eq. (2), we can re-train the classifier as

$$\phi^{**} = \arg\min_{\phi} \mathbb{E}_{(\boldsymbol{x}, y) \sim p_{\mathcal{D}_{\text{CB}}}} \left[ -\log \boldsymbol{p}^{(y)}(\boldsymbol{x}; (\boldsymbol{\theta}^*, \phi)) \right], \tag{3}$$

where $\mathcal{D}_{\text{CB}}$ is the class-balanced training dataset, i.e., the probability of sampling a data $(\boldsymbol{x}, y)$ is given by $p_{\mathcal{D}_{\text{CB}}}((\boldsymbol{x}, y)) = 1/(K \times n_y)$, where $n_y$ is the number of training examples for class $y$. This classifier Re-Training (cRT; Kang et al., 2020) method effectively improves the classification accuracy on tail classes without any changes in the feature extractor $\mathcal{F}_{\boldsymbol{\theta}^*}$.

## 2.2 STOCHASTIC WEIGHT AVERAGING (SWA)

Stochastic Weight Averaging (SWA; Izmailov et al., 2018) is an optimization method to improve generalization performance of deep neural networks. Given a loss function $\mathcal{L}(\Theta)$, the conventional SGD steps towards a local minima by following the gradient direction, where $\eta$ denotes a step size,

$$\Theta_t = \Theta_{t-1} - \eta \nabla_{\Theta} \mathcal{L}(\Theta)|_{\Theta = \Theta_{t-1}}, \tag{4}$$

forming a parameter trajectory $\{\Theta_t\}_{t \geq 1}$. SWA constructs a moving average of parameters for a periodically sampled subset of this trajectory, starting from $\Theta_{\text{SWA}} = \mathbf{0}$,

$$\Theta_{\text{SWA}} = (n\Theta_{\text{SWA}} + \Theta_n)/(n+1), \tag{5}$$

where $\Theta_n$ is usually sampled at the end of every training epochs. This averaging in the weight space implicitly seeks flat minima in the loss surface, and thus enhances generalization. In practice, the averaging phase defined in Eq. (5) starts after the SGD trajectory falls into the basin of the loss function (e.g., after the 75% training epochs), and the learning rate during the averaging phase is set as high values to encourage exploration in the loss surface.

## 2.3 SWA-GAUSSIAN FOR APPROXIMATE BAYESIAN INFERENCE

SWA-Gaussian (SWAG; Maddox et al., 2019) conducts Bayesian inference using a Gaussian approximation to the posterior distribution over the model parameters. With a slight abuse of notation writing the element-wise square as $\Theta^2$, SWAG maintains the second moment for model parameters in addition to the first moment defined in Eq. (5), starting from $\Theta'_{\text{SWA}} = \mathbf{0}$,

$$\Theta'_{\text{SWA}} = (n\Theta'_{\text{SWA}} + \Theta_n^2)/(n+1), \tag{6}$$

to compute a diagonal covariance matrix $\Sigma_{\text{SWAG}} = \text{diag}(\Theta'_{\text{SWA}} - \Theta_{\text{SWA}}^2)$ approximating the sample covariance of parameters captured during SWA. The approximate posterior for the parameters is then constructed as Gaussian, $q(\Theta) = \mathcal{N}(\Theta; \Theta_{\text{SWA}}, \Sigma_{\text{SWAG}})$. As suggested in Maddox et al. (2019), one can also consider a higher-rank approximation for the covariance, but in this paper, we only consider the diagonal approximation.

## 3 LONG-TAILED CLASSIFICATION WITH STOCHASTIC WEIGHT AVERAGING

After the success in supervised image classification tasks (Izmailov et al., 2018), SWA has been further validated for others, including semi-supervised learning (Athiwaratkun et al., 2019) and domain generalization (Cha et al., 2021). In this section, we first study whether the success of SWA continues in the long-tailed classification (Section 3.1). Then we introduce the concept of stochastic representation constructed with SWAG and empirically justify how such stochastic representations can benefit the long-tailed classification (Section 3.2).

**Table 1:** Comparison between SGD and SWA before and after applying cRT on benchmarks. Blue denotes a clear improvement of SWA over SGD, while red denotes deterioration.

| | ImageNet-LT | | | | iNaturalist-2018 | | | |
|---|---|---|---|---|---|---|---|---|
| | Many | Medium | Few | All | Many | Medium | Few | All |
| *Before applying cRT:* | | | | | | | | |
| SGD | $66.84_{\pm 0.26}$ | $40.78_{\pm 0.24}$ | $12.05_{\pm 0.23}$ | $\mathbf{46.91}_{\pm 0.22}$ | $76.31_{\pm 0.52}$ | $67.89_{\pm 0.18}$ | $62.24_{\pm 0.17}$ | $\mathbf{66.52}_{\pm 0.05}$ |
| SWA | $67.71_{\pm 0.11}$ | $40.74_{\pm 0.15}$ | $11.01_{\pm 0.10}$ | $\mathbf{47.08}_{\pm 0.12}$ | $77.26_{\pm 0.25}$ | $68.23_{\pm 0.25}$ | $61.87_{\pm 0.13}$ | $\mathbf{66.65}_{\pm 0.10}$ |
| *After applying cRT:* | | | | | | | | |
| SGD | $62.83_{\pm 0.23}$ | $46.92_{\pm 0.26}$ | $26.33_{\pm 0.16}$ | $50.25_{\pm 0.18}$ | $73.03_{\pm 0.57}$ | $69.09_{\pm 0.10}$ | $66.14_{\pm 0.23}$ | $68.33_{\pm 0.04}$ |
| SWA | $63.54_{\pm 0.18}$ | $47.68_{\pm 0.16}$ | $26.85_{\pm 0.28}$ | $\mathbf{50.95}_{\pm 0.12}$ | $73.30_{\pm 0.73}$ | $69.22_{\pm 0.19}$ | $66.74_{\pm 0.25}$ | $\mathbf{68.66}_{\pm 0.15}$ |

## 3.1 SWA NEEDS THE CLASSIFIER RETRAINING

Following Liu et al. (2019), we report classification accuracy (ACC) on three splits: Many (a set of classes each with over 100 training examples), Medium (a set of classes each with 20-100 training examples), and Few (a set of classes each with under 20 training examples). Table 1 compares SGD and SWA on large-scale benchmarks for long-tailed image classification, including ImageNet-LT and iNaturalist-2018. The experimental results '***Before applying cRT***' displayed in Table 1 show that SWA *does not* bring significant performance gain for long-tailed classification tasks, unlike the previous results on other tasks. Compared to the SGD, 1) SWA does not significantly improve performance, and 2) SWA even degrades performance for the 'Few' split, raising a question about the efficacy of SWA for the long-tailed classification. We diagnose the performance bottleneck of SWA from the perspective of decoupled learning. Specifically, we adopt cRT (Kang et al., 2020) method to verify whether the SWA inherently hinders the quality of the feature extractor. If SWA shows worse performance than SGD even after the classifier re-training, we can conclude that SWA is *not* preferable for representation learning for long-tailed classification. The results '***After applying cRT***' shown in Table 1 demonstrate the classification accuracy for SGD and SWA models after their classifiers are re-trained with Eq. (3). The result shows that SWA improves performance for all splits, indicating that SWA actually enhances the quality of the feature extractor, but the classification layer is acting as a bottleneck as in SGD, and this can be fixed with cRT.

## 3.2 STOCHASTIC REPRESENTATIONS CAPTURE PREDICTIVE UNCERTAINTIES

Now let the feature extractor parameter be constructed from SWA procedure as $\boldsymbol{\theta}_{\text{SWA}}$ and the re-trained classifier parameter as $\boldsymbol{\phi}^*_{\text{SWA}}$, that is,

$$\boldsymbol{\phi}^*_{\text{SWA}} = \arg\min_{\boldsymbol{\phi}} \mathbb{E}_{(\boldsymbol{x},y) \sim p_{\mathcal{D}_{\text{CB}}}} \left[ -\log \boldsymbol{p}^{(y)}(\boldsymbol{x}; (\boldsymbol{\theta}_{\text{SWA}}, \boldsymbol{\phi})) \right]. \tag{7}$$

While the parameters $(\boldsymbol{\theta}_{\text{SWA}}, \boldsymbol{\phi}^*_{\text{SWA}})$ may generalize better than the one obtained from SGD, it is still a point estimate without fully capturing the uncertainty of the predictions. To overcome this limitation, we apply SWAG for the feature extractor parameters $\boldsymbol{\theta}$, approximating the posterior of $\boldsymbol{\theta}$ with a Gaussian distribution $q(\boldsymbol{\theta}|\mathcal{D}) := \mathcal{N}(\boldsymbol{\theta}|\boldsymbol{\theta}_{\text{SWA}}, \boldsymbol{\Sigma}_{\text{SWA}})$. Given $q(\boldsymbol{\theta}|\mathcal{D})$, a predictive distribution for an input $\boldsymbol{x}$ can be approximated as

$$p(y|\boldsymbol{x}, \mathcal{D}; \boldsymbol{\phi}^*_{\text{SWA}}) \approx \int \boldsymbol{p}^{(y)}(\boldsymbol{x}; (\boldsymbol{\theta}, \boldsymbol{\phi}^*_{\text{SWA}})) q(\boldsymbol{\theta}|\mathcal{D}) d\boldsymbol{\theta} \approx \frac{1}{M} \sum_{m=1}^{M} \boldsymbol{p}^{(y)}(\boldsymbol{x}; (\boldsymbol{\theta}_m, \boldsymbol{\phi}^*_{\text{SWA}})), \tag{8}$$

where $\boldsymbol{\theta}_1, \dots, \boldsymbol{\theta}_M \overset{\text{i.i.d.}}{\sim} q(\boldsymbol{\theta}|\mathcal{D})$. Here, we are computing the model average from multiple predictions evaluated with the multiple feature extractor parameters $(\boldsymbol{\theta}_m)_{m=1}^{M}$ to better capture epistemic uncertainty. During that process, we implicitly compute the *stochastic representations* of $\boldsymbol{x}$, that is,

$$\mathcal{F}_m(\boldsymbol{x}) := \mathcal{F}_{\boldsymbol{\theta}_m}(\boldsymbol{x}), \quad \text{for } m = 1, \dots, M. \tag{9}$$

We hypothesize that these stochastic representations reflect the epistemic uncertainty of $\boldsymbol{x}$, so it is important to consider them for the classifier re-training stage. For instance, if $\boldsymbol{x}$ is a hard example (e.g., a sample from tail classes) that is likely to be misclassified, the corresponding stochastic representations are expected to produce predictive distribution having high variances.

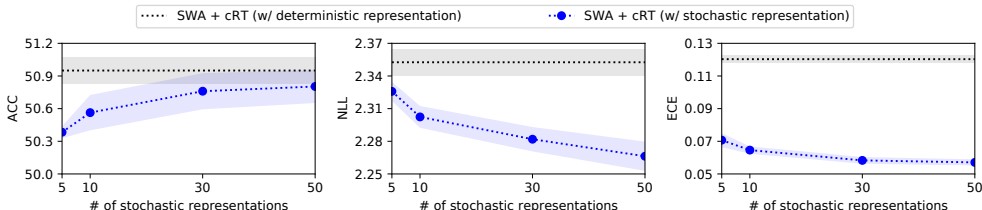

**Figure 2:** Box-and-whisker plots for dispersion values in four different groups of examples. We also compute the Pearson Correlation Coefficient (PCC) value to clarify the positive correlation between negative log-likelihood (NLL) and dispersion.

**Figure 3:** Results for ensemble of stochastic representations over re-trained decision boundaries. Along with classification accuracy (ACC; left), we plot uncertainty estimates, including negative log-likelihood (NLL; center) and expected calibration error (ECE; right). We evaluate models on the test split of ImageNet-LT and colorize standard deviations over four seeds.

**Measuring per-instance dispersion.** We empirically validate our hypothesis on stochastic representations by measuring their dispersion over different inputs. The average cosine distance of the set elements to the set centroid in the $L$-dimensional representation space can quantify such dispersion[1]. To be concrete, we consider the following *per-instance dispersion* for each instance $\boldsymbol{x}$ *in the representation space*,

$$\text{Dispersion}\left(\boldsymbol{x}; \{\boldsymbol{\theta}_m\}_{m=1}^M\right) \coloneqq \frac{1}{M}\sum_{m=1}^M\left[1 - \frac{\overline{\mathcal{F}}(\boldsymbol{x})^\top \mathcal{F}_{\boldsymbol{\theta}_m}(\boldsymbol{x})}{\left\|\overline{\mathcal{F}}(\boldsymbol{x})\right\|_2 \left\|\mathcal{F}_{\boldsymbol{\theta}_m}(\boldsymbol{x})\right\|_2}\right], \tag{10}$$

where $\overline{\mathcal{F}}(\boldsymbol{x}) \coloneqq \sum_{m=1}^M \mathcal{F}_{\boldsymbol{\theta}_m}(\boldsymbol{x})/M$ is the centroid of $M$ stochastic representations $\{\mathcal{F}_{\boldsymbol{\theta}_m}(\boldsymbol{x})\}_{m=1}^M$ for an input $\boldsymbol{x}$. Furthermore, we also measure the Jensen-Shannon Divergence (JSD) for a set of predictions $\{\boldsymbol{p}(\boldsymbol{x}; (\boldsymbol{\theta}_m, \boldsymbol{\phi}_{\text{SWA}}^*))\}_{m=1}^M$,

$$\text{Dispersion}\left(\boldsymbol{x}; \{(\boldsymbol{\theta}_m, \boldsymbol{\phi}_{\text{SWA}^*})\}_{m=1}^M\right) \coloneqq \text{JSD}\left(\{\boldsymbol{p}(\boldsymbol{x}; (\boldsymbol{\theta}_m, \boldsymbol{\phi}_{\text{SWA}}^*))\}_{m=1}^M\right), \tag{11}$$

to quantify the *per-instance dispersion* for each instance $\boldsymbol{x}$ *in the probability space*.

Fig. 2 shows the box-and-whisker plots for dispersion values over 20,000 validation examples from ImageNet-LT. We split them into four disjoint groups consisting of 5,000 instances in each group based on the Negative Log-Likelihood (NLL) values: Q1 (a set of instances having NLLs lower than the first quartile), Q2 (a set of instances having NLLs in the range between the first and second quartiles), Q3 (a set of instances having NLLs in the range between the second and third quartiles), and Q4 (a set of instances having NLLs higher than the third quartile). We confirm that there exists a positive correlation between NLL and dispersion (0.233 for Eq. (10) and 0.385 for Eq. (11)), that is, a hard example (i.e., higher NLL) tends to have more dispersed stochastic representations (i.e., higher dispersion). The higher correlation in Eq. (11) compared to Eq. (10) motivates us to harness the diversity in the probability space instead of the representation space. Empirical results in Section 6.1 will show this indeed leads to improvements. Moreover, we refer the reader to the first paragraph in Appendix B.5 for further analysis of the *per-class dispersion* in the context of long-tailed recognition.

---

[1]We use the cosine distance instead of Euclidean distance since it is a more reasonable choice for the inner-product-based representation space trained with Eq. (1).

**Ensembling predictions with stochastic representations.** We further provide an empirical evidence showing that the uncertainty captured by the stochastic representations can be harnessed by the classifier for robust classification. Specifically, we compute the classification probabilities ensembled over stochastic representations using (8). Note that the classifier parameters $\phi^*_{\text{SWA}}$ is re-trained with $\boldsymbol{\theta}_{\text{SWA}}$ with (7), not with the sampled parameters $(\boldsymbol{\theta}_m)_{m=1}^M$, so in principle, there is no guarantee that $\phi^*_{\text{SWA}}$ is compatible with the stochastic representations computed from $(\boldsymbol{\theta}_m)_{m=1}^M$. Still, if the stochastic representations properly capture the uncertainty of $\boldsymbol{x}$ in the feature space, the ensembled classifier would provide well-calibrated predictions.

Fig. 3 demonstrates how classification accuracy and uncertainty estimates, including NLL and Expected Calibration Error (ECE), vary along with the number of stochastic representations for ensembling defined in Eq. (8). While the classification accuracy remains at a similar level, uncertainty estimates improve as the number of stochastic representations increases, indicating that the stochastic representations indeed captures the uncertainty of inputs that are helpful for robust predictions.

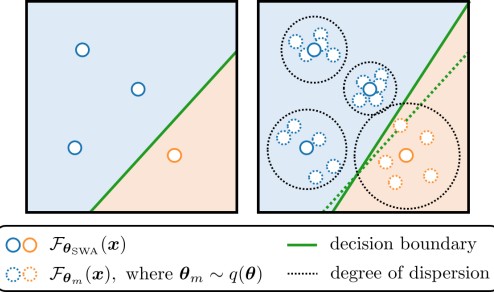

**Figure 4:** A schematic diagram depicting two-dimensional representation space after the first representation learning stage. **Left:** The vanilla approach for the second classifier learning stage re-trains the decision boundary between two classes using deterministic representations. **Right:** Our proposed methods consider stochastic representations having different degrees of dispersion for each input to build more robust decision boundary.

We would like to emphasize that the classifier Eq. (8) is a proof-of-concept model to validate our hypothesis, and there still is a room for improvement. First of all, as we mentioned earlier, the classifier parameters $\phi^*_{\text{SWA}}$ needs to be retrained according to the stochastic representations. Second, even with such re-training, the current form requires multiple forward pass through the feature extractor (with multiple $\boldsymbol{\theta}_m$), incurring undesirably heavy inference cost. In Section 4, we propose a novel re-training algorithm to resolve these issues.

## 4 DECOUPLED LEARNING WITH STOCHASTIC REPRESENTATION

### 4.1 RE-TRAINING CLASSIFIER WITH STOCHASTIC REPRESENTATIONS

A straightforward way to re-train the classifier with stochastic representations would be minimizing

$$\phi^* = \arg\min_{\phi} \mathbb{E}_{(\boldsymbol{x},y)\sim p_{\mathcal{D}_{\text{CB}}}}\left[\mathbb{E}_{\boldsymbol{\theta}\sim q(\boldsymbol{\theta}|\mathcal{D})}[-\log p^{(y)}_{\boldsymbol{\theta},\phi}(\boldsymbol{x})]\right] \approx \arg\min_{\phi} \mathbb{E}_{(\boldsymbol{x},y)\sim p_{\mathcal{D}_{\text{CB}}}}\left[\hat{\mathcal{L}}_{\text{CE}}(\boldsymbol{x},y)\right], \quad (12)$$

where $\boldsymbol{p}^{(y)}_{\boldsymbol{\theta},\phi}(\boldsymbol{x}) := \boldsymbol{p}^{(y)}(\boldsymbol{x};(\boldsymbol{\theta},\phi))$, $\boldsymbol{\theta}_1,\ldots,\boldsymbol{\theta}_M \overset{\text{i.i.d.}}{\sim} q(\boldsymbol{\theta}|\mathcal{D})$, and

$$\hat{\mathcal{L}}_{\text{CE}}(\boldsymbol{x},y) := -\frac{1}{M}\sum_{m=1}^M \log \boldsymbol{p}^{(y)}_{\boldsymbol{\theta}_m,\phi}(\boldsymbol{x}). \quad (13)$$

Fig. 4 depicts a two-dimensional representation space during the classifier re-training stage. While learning from a deterministic feature extractor (i.e., minimizing Eq. (7)) computes the deterministic representations, our proposed method minimizing Eq. (12) builds more robust decision boundaries since it accounts for predictive uncertainties estimated from the stochastic representations.

### 4.2 SELF-DISTILLATION WITH STOCHASTIC REPRESENTATION

While the re-training with (12) helps build a robust classifier, there still is room for improvement. After all, what we are ultimately interested in is the uncertainty in a set of predictions $\{\boldsymbol{p}_{\boldsymbol{\theta}_m,\phi}(\boldsymbol{x})\}_{m=1}^M$, not the stochastic representations $\{\mathcal{F}_{\boldsymbol{\theta}_m}(\boldsymbol{x})\}_{m=1}^M$ themselves. The objective (12) learns the classifier parameters $\phi$ with the *mean* loss (13), so it does not consider the *diversities* of the predictions made

by different stochastic representations. Moreover, as we pointed out earlier, in the current form, we need $M$ forward passes through the feature extractor with $(\boldsymbol{\theta}_m)_{m=1}^M$ to make a prediction. To resolve these issues, we present a self-distillation algorithm that can enhance the robustness of the classifier by transferring diversities in the multiple predictions made with stochastic representations reducing the inference cost to be the same as a single model.

**Setting up teachers and a student.** We first set a set of teachers constructed from the set of stochastic representations. Formally, the $m^{\text{th}}$ teacher $\mathcal{T}_m$ produces a prediction probability for an input $\boldsymbol{x}$ as $\boldsymbol{p}_{\mathcal{T}_m}(\boldsymbol{x}) := \boldsymbol{p}_{\boldsymbol{\theta}_m,\boldsymbol{\phi}}(\boldsymbol{x})$ where $\boldsymbol{\theta}_m \sim q(\boldsymbol{\theta}|\mathcal{D})$. For a student, we first fix the feature extractor parameter as $\boldsymbol{\theta}_{\text{SWA}}$, which is a natural choice considering that the mean of the distribution $q(\boldsymbol{\theta}|\mathcal{D})$ is $\boldsymbol{\theta}_{\text{SWA}}$. The goal is to learn the classifier parameters $\boldsymbol{\phi}$ such that the student prediction $\boldsymbol{p}_{\boldsymbol{\theta}_{\text{SWA}},\boldsymbol{\phi}}(\boldsymbol{x})$ can maximally absorb the diversities in the teacher predictions $(\boldsymbol{p}_{\mathcal{T}_m}(\boldsymbol{x}))_{m=1}^M$.

**Distillation objective.** Following Ryabinin et al. (2021), instead of directly distilling from the *mean* of ensemble predictions $\boldsymbol{p}_{\mathcal{T}}(\boldsymbol{x}) := \sum_m \boldsymbol{p}_{\mathcal{T}_m}(\boldsymbol{x})/M$ as in the original knowledge distillation (Hinton et al., 2015), we distill from the *distribution* of the ensembled predictions. Specifically, we assume that the teacher prediction probabilities $(\boldsymbol{p}_{\mathcal{T}_m}(\boldsymbol{x}))_{m=1}^M$ are Dirichlet distributed with parameter $\boldsymbol{\beta}(\boldsymbol{x})$, that is,

$$\boldsymbol{p}_{\mathcal{T}_1}(\boldsymbol{x}), \dots, \boldsymbol{p}_{\mathcal{T}_M}(\boldsymbol{x}) \overset{\text{i.i.d.}}{\sim} \text{Dir}(\boldsymbol{\beta}(\boldsymbol{x})). \tag{14}$$

The parameter $\boldsymbol{\beta}(\boldsymbol{x})$ can be estimated via the approximate maximum likelihood procedure whose solution is given in a closed-form (Minka, 2000):

$$\boldsymbol{\beta}^{(k)}(\boldsymbol{x}) := \boldsymbol{p}_{\mathcal{T}}^{(k)}(\boldsymbol{x}) \frac{(K-1)/2}{\sum_{j=1}^K \left[ \boldsymbol{p}_{\mathcal{T}}^{(j)}(\boldsymbol{x}) \left( \log \boldsymbol{p}_{\mathcal{T}}^{(j)}(\boldsymbol{x}) - \frac{1}{M} \sum_{m=1}^M \log \boldsymbol{p}_{\mathcal{T}_m}^{(j)}(\boldsymbol{x}) \right) \right]}, \tag{15}$$

for $k = 1, ..., K$. Having computed $\boldsymbol{\beta}(\boldsymbol{x})$, we assume that the output probabilities from a student model (our original model to be re-trained for classifier parameters) are also Dirichlet distributed with parameter $\boldsymbol{\alpha}(\boldsymbol{x})$. We constraint all components in parameters to be greater than 1 (i.e., $\boldsymbol{\beta} \leftarrow \boldsymbol{\beta} + \mathbf{1}$ and $\boldsymbol{\alpha} \leftarrow \boldsymbol{\alpha} + \mathbf{1}$) and minimize the reverse KL divergence $D_{\text{KL}}[\text{Dir}(\boldsymbol{\alpha})\|\text{Dir}(\boldsymbol{\beta})]$ for stable training (Ryabinin et al., 2021). Expanding the KL divergence, we obtain the distillation loss as

$$\hat{\mathcal{L}}_{\text{KD}}(\boldsymbol{x}, y) = \mathbb{E}_{\boldsymbol{p} \sim \text{Dir}(\boldsymbol{\alpha})} \left[ -\sum_{k=1}^K \boldsymbol{p}_{\mathcal{T}}^{(k)}(\boldsymbol{x}) \log \boldsymbol{p}^{(k)} \right] + \frac{1}{\sum_{j=1}^K \boldsymbol{\beta}^{(j)}} D_{\text{KL}} \left[ \text{Dir}(\boldsymbol{\alpha}) \| \text{Dir}(\mathbf{1}) \right], \tag{16}$$

where we compute $\boldsymbol{\alpha}(\boldsymbol{x}) \in \mathbb{R}_+^K$ for $\boldsymbol{\phi} = (\boldsymbol{w}_k, b_k)_{k=1}^K$ as

$$\boldsymbol{\alpha}^{(k)}(\boldsymbol{x}) := \exp\left( \boldsymbol{w}_k^\top \mathcal{F}_{\boldsymbol{\theta}_{\text{SWA}}}(\boldsymbol{x}) + b_k \right) \quad \text{for } k = 1, ..., K. \tag{17}$$

After learning $\boldsymbol{\alpha}(\boldsymbol{x})$, we can take the mean of the student Dirichlet distribution as a representative classification probability for $\boldsymbol{x}$. By the property of the softmax, this actually coincides with the $\boldsymbol{p}_{\boldsymbol{\theta}_{\text{SWA}},\boldsymbol{\phi}}(\boldsymbol{x})$. That is, $\mathbb{E}_{\boldsymbol{p} \sim \text{Dir}(\boldsymbol{\alpha}(\boldsymbol{x}))} \left[ \boldsymbol{p}^{(k)} \right] = \boldsymbol{\alpha}^{(k)}(\boldsymbol{x}) / \sum_{j=1}^K \boldsymbol{\alpha}^{(j)}(\boldsymbol{x}) = \boldsymbol{p}_{\boldsymbol{\theta}_{\text{SWA}},\boldsymbol{\phi}}^{(k)}(\boldsymbol{x})$.

The final version of our algorithm combines the two loss terms $\hat{\mathcal{L}}_{\text{CE}}$ and $\hat{\mathcal{L}}_{\text{KD}}$ for the classifier re-training (see Appendix A.4 for more details). While the first term builds decision boundaries over the stochastic representations, the second term further adjusts decision boundaries by distilling diverse predictions from the stochastic representations. We name this procedure as a *self-distillation* in the sense that the model distills probabilistic outputs within the network itself (Zhang et al., 2019). We empirically validate the effectiveness of the self-distillation in Section 6.

## 5 RELATED WORKS

**Decoupled learning.** Recent works have shown that the performance bottleneck of deep neural classifiers on long-tailed datasets is improper decision boundaries (Kang et al., 2020; Zhang et al., 2021). The vanilla training with instance-balanced sampling gives generalizable representations (Kang et al., 2020), and thus a simple adjusting strategy for the classifier can alleviate such bottleneck. For instance, re-training the classifier from scratch or normalizing the classifier weights with class-balanced sampling (Kang et al., 2020), adjusting the classifier biases with the empirical class frequencies on the training dataset (Menon et al., 2021), and training additional module which performs input-dependent adjustment of the original classifier (Zhang et al., 2021).

**Table 2:** Ablation studies of proposed methods on ImageNet-LT: classification accuracy (ACC), negative log-likelihood (NLL), and expected calibration error (ECE). These results are with Class Balanced Sampling (CBS); refer to Appendix B.1 the results for the other balancing strategies.

| Method | Note | ACC ($\uparrow$) | NLL ($\downarrow$) | ECE ($\downarrow$) |
|---|---|---|---|---|
| SGD w/ classifier re-training | Section 2.1 | $50.25_{\pm 0.18}$ | $2.364_{\pm 0.008}$ | $0.110_{\pm 0.001}$ |
| + (a) introducing SWA for the representation learning | Section 3 | $50.95_{\pm 0.12}$ | $2.353_{\pm 0.012}$ | $0.120_{\pm 0.002}$ |
| + (b) classifier re-training w/ stochastic representation | Section 4.1 | $51.33_{\pm 0.17}$ | $2.340_{\pm 0.012}$ | $0.125_{\pm 0.003}$ |
| + (c) classifier re-training w/ self-distillation | Section 4.2 | $\mathbf{51.66}_{\pm 0.13}$ | $\mathbf{2.203}_{\pm 0.009}$ | $\mathbf{0.074}_{\pm 0.002}$ |

**Knowledge distillation.** The seminal work of Hinton et al. (2015) has shown that distilling knowledge in deep neural networks is an effective way to obtain better generalizing models. One of the common practices in knowledge distillation is employing ensembles as a teacher model (Malinin et al., 2020; Ryabinin et al., 2021) based on the superior performance of the ensemble of deep neural networks (Lakshminarayanan et al., 2017; Ovadia et al., 2019), and several works already applied this to long-tailed recognition (Iscen et al., 2021; Wang et al., 2021). Nevertheless, we would like to clarify that there is a clear difference from them in that our method does not require a costly teacher model (i.e., we do not require an independently trained teacher model).

## 6 EXPERIMENTS

We present extensive experimental results on long-tailed image classification benchmarks for the family of residual networks (He et al., 2016), i.e., ResNet-32 on CIFAR10/100-LT (Cao et al., 2019) and ResNet-50 on ImageNet-LT (Liu et al., 2019) and iNaturalist-2018 (Van Horn et al., 2018). See Appendix C for more details. While some previous literature only reports classification accuracy (ACC), we also provide Negative Log-Likelihood (NLL) and Expected Calibration Error (ECE) that further evaluate the calibration of classification models. See Appendix A.3 for definitions of each metric. Unless specified, we report numbers in Avg.$_{\pm \text{std.}}$ over four random seeds.

### 6.1 ABLATION STUDIES OF PROPOSED METHODS

In this section, we empirically verify that our proposed method progressively improves upon the previous baseline. More precisely, we test the following step-by-step; (a) introducing SWA for the representation learning from Section 3, (b) introducing stochastic representation for the classifier re-training from Section 4.1, and (c) introducing self-distillation strategy from Section 4.2.

Here, we consider the following balancing strategies during the classifier re-training; 1) Class-Balanced Sampling (CBS; Kang et al., 2020), 2) Generalized Re-Weighting (GRW; Zhang et al., 2021), and 3) Logit Adjustment (LA; Menon et al., 2021). Such strategies to overcome the difficulty of imbalanced data distribution are essential to re-train proper decision boundaries in the decoupled learning scheme. Refer to Appendix A.1 for more details on balancing strategies.

Table 2 shows the evaluation results for each step on ImageNet-LT when we use CBS. To summarize, (a) the results displayed in the first and second rows show that SWA improves the SGD baseline with a decoupling scheme, as we discussed before in Section 3.1. (b) Moreover, the results displayed in the second and third rows show that using stochastic representations to re-train the classifier improves the classification accuracy. It indicates that the stochastic representations contribute to building more robust decision boundaries, as depicted in Fig. 4. However, our experimental results up to this point show that the improvements in uncertainty estimates are not significant. (c) Finally, our proposed approach in Section 4.2 significantly outperforms previous baselines for every metric we measured. Notably, the improvement consistently appears for all balancing strategies we considered (see Appendix B.1 for the full results). In particular, definite improvement in uncertainty estimates confirms the effectiveness of the proposed self-distillation method.

### 6.2 RESULTS ON IMAGE CLASSIFICATION TASKS

We compare our approach to the existing methods for long-tailed learning; cRT (Kang et al., 2020), LWS (Kang et al., 2020), LA (Menon et al., 2021), and DisAlign (Zhang et al., 2021). See Appendices A.1 and A.2 for more details on existing methods. Since the representation learning using

**Table 3:** Results on ImageNet-LT and iNaturalist-2018: classification accuracy (ACC), negative log-likelihood (NLL), and expected calibration error (ECE). Full results are available in Appendix B.1.

| Method | ImageNet-LT | | | iNaturalist-2018 | | |
|---|---|---|---|---|---|---|
| | ACC (↑) | NLL (↓) | ECE (↓) | ACC (↑) | NLL (↓) | ECE (↓) |
| SWA | $47.08_{\pm 0.12}$ | $2.631_{\pm 0.009}$ | $0.187_{\pm 0.002}$ | $66.65_{\pm 0.10}$ | $1.568_{\pm 0.005}$ | $0.071_{\pm 0.001}$ |
| + cRT (Kang et al., 2020) | $50.95_{\pm 0.12}$ | $2.353_{\pm 0.012}$ | $0.120_{\pm 0.002}$ | $68.66_{\pm 0.15}$ | $1.546_{\pm 0.002}$ | $0.061_{\pm 0.002}$ |
| + LWS (Kang et al., 2020) | $51.60_{\pm 0.10}$ | $2.189_{\pm 0.007}$ | $0.077_{\pm 0.002}$ | $70.53_{\pm 0.09}$ | $1.370_{\pm 0.002}$ | $0.049_{\pm 0.001}$ |
| + LA (Menon et al., 2021) | $51.62_{\pm 0.05}$ | $2.206_{\pm 0.009}$ | $0.077_{\pm 0.002}$ | $69.63_{\pm 0.20}$ | $1.466_{\pm 0.003}$ | $0.038_{\pm 0.001}$ |
| + DisAlign (Zhang et al., 2021) | $\mathbf{52.18}_{\pm 0.11}$ | $2.673_{\pm 0.014}$ | $0.215_{\pm 0.002}$ | $\mathbf{70.81}_{\pm 0.10}$ | $1.410_{\pm 0.003}$ | $0.076_{\pm 0.000}$ |
| **+ SRepr (ours)** | $\mathbf{52.12}_{\pm 0.06}$ | $\mathbf{2.130}_{\pm 0.006}$ | $\mathbf{0.037}_{\pm 0.001}$ | $70.79_{\pm 0.17}$ | $\mathbf{1.353}_{\pm 0.002}$ | $\mathbf{0.036}_{\pm 0.002}$ |

**Table 4:** Further comparisons in classification accuracy (ACC) with state-of-the-art methods on ImageNet-LT. Results for the baselines came from the corresponding paper. [†]This actually requires training cost of 600 epochs since rwSAM doubles forward and backward passes for each iteration.

| Method | Epoch | Network | Training details | ACC (↑) |
|---|---|---|---|---|
| KCL (Kang et al., 2021) | 200 | ResNet-50 | - | 51.5 |
| **SWA + SRepr (ours)** | 200 | ResNet-50 | - | **53.8** |
| MiSLAS (Zhong et al., 2021) | 180 | ResNet-50 | mixup (Zhang et al., 2018) | 52.7 |
| **SWA + SRepr (ours)** | 180 | ResNet-50 | mixup (Zhang et al., 2018) | **53.9** |
| LADE (Hong et al., 2021) | 180 | ResNeXt-50 (Xie et al., 2017) | - | 53.0 |
| **SWA + SRepr (ours)** | 180 | ResNeXt-50 (Xie et al., 2017) | - | **54.6** |
| Liu et al. (2021) | 300 | ResNet-50 | MoCo v2 (He et al., 2020) | 55.0 |
| **SWA + SRepr (ours)** | 300 | ResNet-50 | mixup (Zhang et al., 2018) | **55.1** |
| Liu et al. (2021) | 300[†] | ResNet-50 | MoCo v2 (He et al., 2020) + rwSAM (Foret et al., 2021) | 55.5 |
| **SWA + SRepr (ours)** | 400 | ResNet-50 | mixup (Zhang et al., 2018) | **55.7** |

SWA, we introduced in Section 3, is also compatible with previous classifier re-training methods, we hereby report the results built upon this for a fair comparison.

Table 3 presents the evaluation results on ImageNet-LT and iNaturalist-2018. **SWA + SRepr** denotes the final version of our proposed method, previously discussed in Section 6.1, i.e., self-distillation using stochastic representation and balanced by logit adjustment. It demonstrates that the proposed SRepr outperforms baselines, even if existing methods strengthen with SWA. While DisAlign, improved by our proposed SWA, achieves competitive accuracy, SRepr still shows better uncertainty estimates. Refer to Appendix B.1 for a full table, including classification accuracy for each split (i.e., Many, Medium, and Few) and baseline results without applying SWA. Appendix B.2 also provides the results on CIFAR10/100-LT, which show the same tendency. Besides, Appendix B.5 provides further analysis of ours, which validates the effectiveness of the proposed method.

**Further comparisons with state-of-the-art methods.** One can claim the performance gap between ours and the state-of-the-art methods. This issue appears due to the shorter training epochs (i.e., 100 training epochs) and the vanilla data augmentation strategy (i.e., random resized cropping and horizontal flipping) in our experimental setups. Table 4 further clarifies that our proposed approach outperforms the state-of-the-art methods (Kang et al., 2021; Zhong et al., 2021; Hong et al., 2021; Liu et al., 2021) when the training costs get in line (see Appendix B.3 for more details). Moreover, Appendix B.4 further verifies the compatibility between ours and the existing state-of-the-art methods (Wang et al., 2021; Park et al., 2022; Zhang et al., 2022; Zhu et al., 2022), which clarifies ours can be combined with existing algorithms taking different approaches.

## 7 CONCLUSION

In this paper, we proposed a simple yet effective classifier re-training strategy for long-tailed learning in decoupled learning scheme. We first showed that successful representation learning is achievable by SWA without any complex training methods. To the best of our knowledge, this is the first attempt to introduce SWA into long-tailed learning. While just combining existing classifier re-training methods with the representation learned by SWA shows better performance than with the vanilla SGD, we further proposed a novel self-distillation strategy that significantly improves uncertainty estimates of the final classifier.

## ETHIC AND REPRODUCIBILITY STATEMENTS

For societal impacts, there can be potential negative impacts associated with face recognition (e.g., privacy threats) since face data often exhibit long-tailed distribution over entities (Zhang et al., 2017). For reproducibility, Appendix C contains all the experimental setup including datasets and hyperparameters.

## ACKNOWLEDGEMENTS

This work was partly supported by Institute of Information & Communications Technology Planning & Evaluation (IITP) grant funded by the Korea government (MSIT) (No. 2019-0-00075, Artificial Intelligence Graduate School Program (KAIST)), Samsung Electronics Co., Ltd (IO201214-08176-01), and the National Research Foundation of Korea (NRF) grant funded by the Korea government (MSIT) (No. 2022R1A5A708390811).

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

# A    SUPPLEMENTARY MATERIALS

## A.1    BALANCING STRATEGIES FOR RE-TRAINING CLASSIFIER

In Section 6.1, we considered the following balancing strategies for the classifier re-training stage. Such strategies for re-balancing the training data having long-tailed distribution over classes is essential to construct proper decision boundaries over the representation space.

- **Class-Balanced Sampling (CBS)** is the most straightforward way to re-balance long-tailed data distribution by sampling. The probability of sampling a training example from class $k$ is $1/K$, where $K$ is the number of training classes. More precisely, the softmax cross-entropy loss over the class-balanced training dataset $\mathcal{D}_{\mathrm{CB}}$ is

$$\mathbb{E}_{(\boldsymbol{x},y)\sim p_{\mathcal{D}_{\mathrm{CB}}}}\left[-\log\frac{\exp\left(\boldsymbol{w}_y^\top\mathcal{F}_{\boldsymbol{\theta}}(\boldsymbol{x})+b_y\right)}{\sum_{j=1}^{K}\exp\left(\boldsymbol{w}_j^\top\mathcal{F}_{\boldsymbol{\theta}}(\boldsymbol{x})+b_j\right)}\right], \tag{18}$$

where the probability of sampling a data $(\boldsymbol{x},y)$ is given by $p_{\mathcal{D}_{\mathrm{CB}}}((\boldsymbol{x},y))=1/(K\times n_y)$ and $n_y$ denotes the number of training examples for class $y$.

- **Generalized Re-Weighting (GRW)** performs loss re-weighting using the empirical class frequencies $\{\pi_k\}_{k=1}^{K}$ on the training dataset (Zhang et al., 2021). More precisely, the re-weighted version of the softmax cross-entropy loss over the training dataset $\mathcal{D}$ is

$$\mathbb{E}_{(\boldsymbol{x},y)\sim\mathcal{D}}\left[-\frac{(1/\pi_y)^\rho}{\sum_{j=1}^{K}(1/\pi_j)^\rho}\log\frac{\exp\left(\boldsymbol{w}_y^\top\mathcal{F}_{\boldsymbol{\theta}}(\boldsymbol{x})+b_y\right)}{\sum_{j=1}^{K}\exp\left(\boldsymbol{w}_j^\top\mathcal{F}_{\boldsymbol{\theta}}(\boldsymbol{x})+b_j\right)}\right], \tag{19}$$

where $\rho\geq 0$ is a hyper-parameter that controls the scale of per-class weighting coefficients, e.g., it reduces to the instance-balanced re-weighting with $\rho=0.0$, and to the class-balanced re-weighting with $\rho=1.0$. We use $\rho=1.0$ throughout the experiments.

- **Logit Adjustment (LA)** is a simple but efficient way to minimize the balanced error (i.e., average of per-class error rates) using the empirical class frequencies $\{\pi_k\}_{k=1}^{K}$ on the training dataset (Menon et al., 2021). More precisely, the logit adjusted version of the softmax cross-entropy loss over the training dataset $\mathcal{D}$ is

$$\mathbb{E}_{(\boldsymbol{x},y)\sim\mathcal{D}}\left[-\log\frac{\exp\left(\boldsymbol{w}_y^\top\mathcal{F}_{\boldsymbol{\theta}}(\boldsymbol{x})+b_y+\rho\log\pi_y\right)}{\sum_{j=1}^{K}\exp\left(\boldsymbol{w}_j^\top\mathcal{F}_{\boldsymbol{\theta}}(\boldsymbol{x})+b_j+\rho\log\pi_j\right)}\right], \tag{20}$$

where $\rho\geq 0$ is a hyper-parameter that controls the scale of offset to each of the logits. We use $\rho=1.0$ throughout the experiments.

## A.2    CLASSIFIER RE-TRAINING METHODS

Assume that we have pre-trained parameters $\boldsymbol{\Theta}^*=(\boldsymbol{\theta}^*,\boldsymbol{\phi}^*)$ after the first representation learning stage. From this, the classifier re-training methods aim to find a new classifier parameters $\boldsymbol{\phi}^{**}=(\boldsymbol{w}_k^{**},b_k^{**})_{k=1}^{K}$, while the feature extractor parameters $\boldsymbol{\theta}^*$ is frozen.

- **Classifier Re-Training (cRT)** is the most straightforward way to re-train the classifier from scratch (Kang et al., 2020). Specifically, it first randomly re-initializes the classifier parameters $\boldsymbol{\phi}$ and re-trains them on the class-balanced training dataset $\mathcal{D}_{\mathrm{CB}}$,

$$\boldsymbol{\phi}^{**}=\arg\min_{\boldsymbol{\phi}}\mathbb{E}_{(\boldsymbol{x},y)\sim p_{\mathcal{D}_{\mathrm{CB}}}}\left[-\log\boldsymbol{p}^{(y)}(\boldsymbol{x};(\boldsymbol{\theta}^*,\boldsymbol{\phi}))\right]. \tag{21}$$

- **Learnable Weight Scaling (LWS)** only re-trains the scale of pre-trained weights $(\boldsymbol{w}_k^*)_{k=1}^{K}$ while the direction is kept (Kang et al., 2020). Specifically, it introduces trainable parameter $\tau\in\mathbb{R}$ which controls the intensity of the weight normalization,

$$\boldsymbol{\phi}(\tau)=(\boldsymbol{w}_k^*/\|\boldsymbol{w}_k^*\|^\tau,b_k^*)_{k=1}^{K}. \tag{22}$$

and finds $\boldsymbol{\phi}^{**}=\boldsymbol{\phi}(\tau^*)$ on the class-balanced training dataset $\mathcal{D}_{\mathrm{CB}}$, where

$$\tau^*=\arg\min_{\tau}\mathbb{E}_{(\boldsymbol{x},y)\sim p_{\mathcal{D}_{\mathrm{CB}}}}\left[-\log\boldsymbol{p}^{(y)}(\boldsymbol{x};(\boldsymbol{\theta}^*,\boldsymbol{\phi}(\tau)))\right]. \tag{23}$$

- **Distribution Alignment (DisAlign)** trains additional modules while the original classifier parameters $\phi^*$ is kept (Zhang et al., 2021). Specifically, it introduces trainable parameters $\{\alpha_k \in \mathbb{R}, \beta_k \in \mathbb{R}\}_{k=1}^K \cup \{\boldsymbol{\gamma} \in \mathbb{R}^K, \delta \in \mathbb{R}\}$ which performs a distribution calibration,

$$z_k(\boldsymbol{x}) \leftarrow \boldsymbol{w}_k^{*\top} \mathcal{F}_{\boldsymbol{\theta}^*}(\boldsymbol{x}) + b_k^*, \tag{24}$$

$$\sigma(\boldsymbol{x}) \leftarrow \mathrm{Sigmoid}(\boldsymbol{\gamma}^\top z_k(\boldsymbol{x}) + \delta), \tag{25}$$

$$\hat{z}_k(\boldsymbol{x}) \leftarrow \sigma(\boldsymbol{x}) \cdot (\alpha_k z_k(\boldsymbol{x}) + \beta_k) + (1 - \sigma(\boldsymbol{x})) \cdot z_k(\boldsymbol{x}), \quad \text{for } k = 1, ..., K, \tag{26}$$

  where $z_k$ and $\hat{z}_k$ respectively denote the original logits and the calibrated logits. While both $\boldsymbol{\theta}^*$ and $\phi^*$ are fixed, it finds the optimal value of $\{\alpha_k, \beta_k\}_{k=1}^K \cup \{\boldsymbol{\gamma}, \delta\} = \{\alpha_k^*, \beta_k^*\}_{k=1}^K \cup \{\boldsymbol{\gamma}^*, \delta^*\}$ on the training dataset $\mathcal{D}$ with Generalized Re-Weighting (GRW).

## A.3 EVALUATION METRICS

The problem we addressed in this paper is the $K$-way classification problem. Let $\mathcal{P} : \boldsymbol{x} \mapsto \boldsymbol{p}$ be a model that outputs a categorical probability $\boldsymbol{p} \in [0,1]^K$ for a given input $\boldsymbol{x}$. Following metrics are reported for all of the methods using our implementation.

- **Accuracy (ACC; higher is better):**

$$\mathrm{ACC}(\mathcal{P}, \mathcal{D}) = \mathbb{E}_{(\boldsymbol{x},y)\in\mathcal{D}} \left[ \left[ y = \arg\max_k \mathcal{P}^{(k)}(\boldsymbol{x}) \right] \right], \tag{27}$$

  where inner $[\cdot]$ denotes the Iverson bracket.

- **Negative log-likelihood (NLL; lower is better):**

$$\mathrm{NLL}(\mathcal{P}, \mathcal{D}) = \mathbb{E}_{(\boldsymbol{x},y)\in\mathcal{D}} \left[ -\log \mathcal{P}^{(y)}(\boldsymbol{x}) \right], \tag{28}$$

  which is equivalent to the softmax cross-entropy loss used in training.

- **Expected calibration error (ECE; lower is better):**

$$\mathrm{ECE}(\mathcal{P}, \mathcal{D}; N) = \sum_{n=1}^{N} \delta_n \left( |\mathcal{B}_n|/|\mathcal{D}| \right), \tag{29}$$

  where $\{\mathcal{B}_1, ..., \mathcal{B}_N\}$ is a partition of $\mathcal{D}$,

$$\mathcal{B}_n = \left\{ (\boldsymbol{x}, y) \in \mathcal{D} \mid \max_k \mathcal{P}^{(k)}(\boldsymbol{x}) \in \left( \frac{n-1}{N}, \frac{n}{N} \right] \right\}, \quad \text{for } n = 1, ..., N, \tag{30}$$

  and $\delta_n$ denotes a calibration error for the $n$th bin $\mathcal{B}_n$,

$$\delta_n = \left| \mathrm{ACC}(\mathcal{P}, \mathcal{B}_n) - \mathbb{E}_{(\boldsymbol{x},\cdot)\in\mathcal{B}_n} \left[ \max_k \mathcal{P}^{(k)}(\boldsymbol{x}) \right] \right|, \quad \text{for } n = 1, ..., N. \tag{31}$$

  We used $N = 15$ bins in this paper.

## A.4 DETAILED IMPLEMENTATION OF THE PROPOSED METHOD

**Loss function** During the optimization, we simply fix $M = 10$ and use equal weights for two terms in the final objective, i.e., the actual implementation is

$$\underbrace{0.5\hat{\mathcal{L}}_{\mathrm{CE}}}_{\substack{\text{cross-entropy} \\ \text{term}}} + \underbrace{0.5\hat{\mathcal{L}}_{\mathrm{KD}}}_{\substack{\text{self-distillation} \\ \text{term}}} . \tag{32}$$

Also, we blocked the gradient flow through the target Dirichlet distribution in the self-distillation term, i.e., `jax.lax.stop_gradient(`$\boldsymbol{\beta}$`)` in JAX library. Our training diverges when using the typical softmax outputs due to the sharp target Dirichlet distribution. Thus, we use the following *temperature-scaled* softmax outputs to stabilize the optimization on the self-distillation term $\hat{\mathcal{L}}_{\mathrm{KD}}$,

$$\boldsymbol{p}_{\boldsymbol{\theta}}^{(k)}(\boldsymbol{x}) \leftarrow \frac{\exp\left( (\boldsymbol{w}_k^\top \mathcal{F}_{\boldsymbol{\theta}}^{(k)}(\boldsymbol{x}) + b_k)/\tau_{\mathrm{KD}} \right)}{\sum_{j=1}^K \exp\left( (\boldsymbol{w}_j^\top \mathcal{F}_{\boldsymbol{\theta}}^{(j)}(\boldsymbol{x}) + b_j)/\tau_{\mathrm{KD}} \right)}, \quad \text{for } k = 1, ..., K. \tag{33}$$

Table 5 shows results for the final version of our algorithm (i.e., SRepr in Section 6.2) on the validation split of ImageNet-LT swept over $\tau_{\mathrm{KD}} \in \{1, 2, 5, 10, 20\}$. Training losses were unstable for $\tau \in \{1, 2, 5\}$ and thus we fix $\tau_{\mathrm{KD}} = 20$ throughout all experiments.

**Table 5:** Validation results of SRepr with varying temperatures (i.e., $\tau_{KD}$ in Appendix A.4) on ImageNet-LT. 'N/A' denotes the training diverges.

| ImageNet-LT (Val) | ACC (↑) | | | | NLL (↓) | ECE (↓) |
|---|---|---|---|---|---|---|
| | Many | Medium | Few | All | | |
| $\tau_{KD} = 1.0$ | N/A | N/A | N/A | N/A | N/A | N/A |
| $\tau_{KD} = 2.0$ | N/A | N/A | N/A | N/A | N/A | N/A |
| $\tau_{KD} = 5.0$ | $66.86_{\pm0.10}$ | $41.58_{\pm0.09}$ | $7.48_{\pm0.09}$ | $46.67_{\pm0.05}$ | $4.238_{\pm0.009}$ | $0.294_{\pm0.001}$ |
| $\tau_{KD} = 10.0$ | $64.21_{\pm0.22}$ | $50.51_{\pm0.14}$ | $32.70_{\pm0.38}$ | $53.36_{\pm0.09}$ | $2.032_{\pm0.001}$ | $0.034_{\pm0.000}$ |
| $\tau_{KD} = 20.0$ | $64.20_{\pm0.18}$ | $50.52_{\pm0.14}$ | $32.89_{\pm0.49}$ | $53.40_{\pm0.18}$ | $2.059_{\pm0.049}$ | $0.030_{\pm0.005}$ |

**Table 6:** Ablation study of proposed methods with various balancing strategies on ImageNet-LT: classification accuracy (ACC), negative log-likelihood (NLL), and expected calibration error (ECE).

| Balancing Strategy | Proposed Methods | | | ACC (↑) | NLL (↓) | ECE (↓) |
|---|---|---|---|---|---|---|
| | Repr. learning with SWA (Section 3) | Classifier learning with stochastic repr. (Section 4.1) | Classifier learning with self-distillation (Section 4.2) | | | |
| *Without re-training classifier* | | | | $46.91_{\pm0.22}$ | $\mathbf{2.546}_{\pm0.009}$ | $\mathbf{0.158}_{\pm0.003}$ |
| | ✓ | | | $47.08_{\pm0.12}$ | $2.631_{\pm0.009}$ | $0.187_{\pm0.002}$ |
| CBS (Kang et al., 2020) | ✓ | | | $50.25_{\pm0.18}$ | $2.364_{\pm0.008}$ | $0.110_{\pm0.001}$ |
| | ✓ | | | $50.95_{\pm0.12}$ | $2.353_{\pm0.012}$ | $0.120_{\pm0.002}$ |
| | ✓ | ✓ | | $51.33_{\pm0.17}$ | $2.340_{\pm0.012}$ | $0.125_{\pm0.003}$ |
| | ✓ | ✓ | ✓ | $\mathbf{51.66}_{\pm0.13}$ | $\mathbf{2.203}_{\pm0.009}$ | $\mathbf{0.074}_{\pm0.002}$ |
| GRW (Zhang et al., 2021) | ✓ | | | $50.77_{\pm0.13}$ | $2.243_{\pm0.007}$ | $0.026_{\pm0.001}$ |
| | ✓ | | | $51.33_{\pm0.16}$ | $2.220_{\pm0.010}$ | $0.041_{\pm0.002}$ |
| | ✓ | ✓ | | $51.73_{\pm0.11}$ | $2.206_{\pm0.008}$ | $0.056_{\pm0.003}$ |
| | ✓ | ✓ | ✓ | $\mathbf{52.08}_{\pm0.04}$ | $\mathbf{2.133}_{\pm0.005}$ | $\mathbf{0.019}_{\pm0.001}$ |
| LA (Menon et al., 2021) | ✓ | | | $50.97_{\pm0.13}$ | $2.231_{\pm0.004}$ | $0.063_{\pm0.001}$ |
| | ✓ | | | $51.62_{\pm0.05}$ | $2.206_{\pm0.009}$ | $0.077_{\pm0.002}$ |
| | ✓ | ✓ | | $51.84_{\pm0.18}$ | $2.208_{\pm0.009}$ | $0.090_{\pm0.003}$ |
| | ✓ | ✓ | ✓ | $\mathbf{52.12}_{\pm0.06}$ | $\mathbf{2.130}_{\pm0.006}$ | $\mathbf{0.037}_{\pm0.001}$ |

**Pseudo-code for the proposed method** Algorithm 1 summarizes the proposed method in pseudo-code. Note that training SWA-Gaussian with a diagonal covariance has virtually no additional cost over conventional training with SGD. It only requires extra space for storing the first and second moments of the backbone parameters (i.e., lines 1-10). Instead, an additional training cost of our approach compared with the existing methods (Kang et al., 2020; Zhang et al., 2021) come from multiple forward passes of the backbone network during the classifier re-training stage (i.e., lines 11-17). Please refer to Appendix B.5 for further investigation regarding this issue.

# B    ADDITIONAL EXPERIMENTS

## B.1    FULL TABLE OF RESULTS

**Ablation studies of proposed methods (Section 6.1).** Table 6 is an extended version of Table 2. It provides the results when we apply the following balancing strategies; CBS, GRW, and LA. The arguments we discussed in Section 6.1 consistently hold for all balancing strategies we considered.

**Results on image classification tasks (Section 6.2).** Table 7 is an extended version of Table 3. Here, we also provide detailed classification accuracy on three splits introduced in (Liu et al., 2019): Many (a set of classes each with over 100 training examples), Medium (a set of classes each with 20-100 training examples), and Few (a set of classes each with under 20 training examples).

## B.2    ADDITIONAL RESULTS ON CIFAR10/100-LT

We also provide the experimental results on CIFAR10-LT and CIFAR100-LT. Table 8 shows that our approach outperforms the baselines in terms of every metric we measured. It clearly demonstrates that the proposed approach consistently benefits from training robust decision boundaries regardless of the scale of datasets.

**Table 7:** Full restuls on ImageNet-LT and iNaturalist-2018: classification accuracy (ACC), negative log-likelihood (NLL), and expected calibration error (ECE).

| ImageNet-LT | ACC (↑) | | | | NLL (↓) | ECE (↓) |
|---|---|---|---|---|---|---|
| | Many | Medium | Few | All | | |
| SGD | $66.84_{\pm0.26}$ | $40.78_{\pm0.24}$ | $12.05_{\pm0.23}$ | $46.91_{\pm0.22}$ | $2.546_{\pm0.009}$ | $0.158_{\pm0.003}$ |
| + cRT (Kang et al., 2020) | $62.83_{\pm0.23}$ | $46.92_{\pm0.26}$ | $26.33_{\pm0.16}$ | $50.25_{\pm0.18}$ | $2.364_{\pm0.008}$ | $0.110_{\pm0.001}$ |
| + LWS (Kang et al., 2020) | $63.23_{\pm0.26}$ | $47.57_{\pm0.24}$ | $27.78_{\pm0.23}$ | $50.91_{\pm0.15}$ | $\mathbf{2.197}_{\pm0.007}$ | $\mathbf{0.054}_{\pm0.001}$ |
| + LA (Menon et al., 2021) | $60.79_{\pm0.20}$ | $48.11_{\pm0.14}$ | $33.20_{\pm0.34}$ | $50.97_{\pm0.13}$ | $2.231_{\pm0.004}$ | $0.063_{\pm0.001}$ |
| + DisAlign (Zhang et al., 2021) | $61.63_{\pm0.39}$ | $48.68_{\pm0.11}$ | $32.71_{\pm0.45}$ | $\mathbf{51.49}_{\pm0.15}$ | $2.596_{\pm0.012}$ | $0.202_{\pm0.002}$ |
| SWA (ours) | $67.71_{\pm0.11}$ | $40.74_{\pm0.15}$ | $11.01_{\pm0.10}$ | $47.08_{\pm0.12}$ | $2.631_{\pm0.009}$ | $0.187_{\pm0.002}$ |
| + cRT (Kang et al., 2020) | $63.54_{\pm0.18}$ | $47.68_{\pm0.16}$ | $26.85_{\pm0.28}$ | $50.95_{\pm0.12}$ | $2.353_{\pm0.012}$ | $0.120_{\pm0.002}$ |
| + LWS (Kang et al., 2020) | $63.51_{\pm0.30}$ | $48.53_{\pm0.07}$ | $28.66_{\pm0.45}$ | $51.60_{\pm0.10}$ | $2.189_{\pm0.007}$ | $0.077_{\pm0.002}$ |
| + LA (Menon et al., 2021) | $61.60_{\pm0.07}$ | $48.70_{\pm0.03}$ | $33.68_{\pm0.34}$ | $51.62_{\pm0.05}$ | $2.206_{\pm0.009}$ | $0.077_{\pm0.002}$ |
| + DisAlign (Zhang et al., 2021) | $62.43_{\pm0.20}$ | $49.48_{\pm0.15}$ | $32.65_{\pm0.43}$ | $\mathbf{52.18}_{\pm0.11}$ | $2.673_{\pm0.014}$ | $0.215_{\pm0.002}$ |
| + SRepr (ours) | $62.52_{\pm0.26}$ | $49.44_{\pm0.18}$ | $32.14_{\pm0.41}$ | $\mathbf{52.12}_{\pm0.06}$ | $\mathbf{2.130}_{\pm0.006}$ | $\mathbf{0.037}_{\pm0.001}$ |

| iNaturalist-2018 | ACC (↑) | | | | NLL (↓) | ECE (↓) |
|---|---|---|---|---|---|---|
| | Many | Medium | Few | All | | |
| SGD | $76.31_{\pm0.52}$ | $67.89_{\pm0.18}$ | $62.24_{\pm0.17}$ | $66.52_{\pm0.05}$ | $1.568_{\pm0.006}$ | $0.048_{\pm0.003}$ |
| + cRT (Kang et al., 2020) | $73.03_{\pm0.57}$ | $69.09_{\pm0.10}$ | $66.14_{\pm0.23}$ | $68.33_{\pm0.04}$ | $1.537_{\pm0.006}$ | $0.037_{\pm0.002}$ |
| + LWS (Kang et al., 2020) | $72.35_{\pm0.43}$ | $70.11_{\pm0.34}$ | $69.73_{\pm0.40}$ | $70.19_{\pm0.08}$ | $\mathbf{1.386}_{\pm0.006}$ | $0.030_{\pm0.001}$ |
| + LA (Menon et al., 2021) | $69.90_{\pm0.50}$ | $69.34_{\pm0.14}$ | $69.66_{\pm0.19}$ | $69.49_{\pm0.15}$ | $1.477_{\pm0.005}$ | $\mathbf{0.015}_{\pm0.002}$ |
| + DisAlign (Zhang et al., 2021) | $71.68_{\pm0.31}$ | $70.73_{\pm0.25}$ | $69.51_{\pm0.39}$ | $\mathbf{70.35}_{\pm0.21}$ | $1.428_{\pm0.006}$ | $0.064_{\pm0.001}$ |
| SWA (ours) | $77.26_{\pm0.25}$ | $68.23_{\pm0.25}$ | $61.87_{\pm0.13}$ | $66.65_{\pm0.10}$ | $1.568_{\pm0.005}$ | $0.071_{\pm0.001}$ |
| + cRT (Kang et al., 2020) | $73.30_{\pm0.73}$ | $69.22_{\pm0.19}$ | $66.74_{\pm0.25}$ | $68.66_{\pm0.15}$ | $1.546_{\pm0.002}$ | $0.061_{\pm0.002}$ |
| + LWS (Kang et al., 2020) | $72.82_{\pm0.45}$ | $70.43_{\pm0.25}$ | $70.06_{\pm0.15}$ | $70.53_{\pm0.09}$ | $1.370_{\pm0.002}$ | $0.049_{\pm0.001}$ |
| + LA (Menon et al., 2021) | $69.70_{\pm0.67}$ | $69.47_{\pm0.11}$ | $69.82_{\pm0.68}$ | $69.63_{\pm0.20}$ | $1.466_{\pm0.003}$ | $0.038_{\pm0.001}$ |
| + DisAlign (Zhang et al., 2021) | $72.34_{\pm0.57}$ | $71.27_{\pm0.10}$ | $69.84_{\pm0.18}$ | $\mathbf{70.81}_{\pm0.10}$ | $1.410_{\pm0.003}$ | $0.076_{\pm0.000}$ |
| + SRepr (ours) | $70.70_{\pm0.31}$ | $70.83_{\pm0.20}$ | $70.76_{\pm0.32}$ | $\mathbf{70.79}_{\pm0.17}$ | $\mathbf{1.353}_{\pm0.002}$ | $\mathbf{0.036}_{\pm0.002}$ |

### B.3 FURTHER COMPARISONS WITH STATE-OF-THE-ART METHODS

**Increasing the number of training epochs.** Throughout the main text, we trained all the competing methods for 100 training epochs on ImageNet-LT, which is sufficient to validate the efficacy of our approach. However, state-of-the-art performances reported in other papers are typically from long training epochs than we set in our experiments. We thus provided the results when the number of training epochs gets in line in Table 4. It clearly demonstrates that ours outperforms baselines by a wide margin. Besides, we also tested our method for ResNeXt-50 architecture (Xie et al., 2017) for a fair comparison with LADE (Hong et al., 2021).

**Applying mixup augmentation.** Zhong et al. (2021) employed mixup augmentation (Zhang et al., 2018) in decoupled training, which is actually one of the main ingredients to achieving that level of performance they reported. However, the mixup augmentation is not exclusive to their method but applicable to generic approaches, including ours. In Table 4, our approach enhanced by the mixup augmentation indeed outperforms the previous baseline utilizing the mixup (Zhong et al., 2021).

### B.4 COMBINING OURS WITH THE EXISTING STATE-OF-THE-ART METHODS

Apart from the decoupled learning scheme (Kang et al., 2020) we mainly considered in the main text, there are other groups of methods achieving state-of-the-art performances: a) utilizing multiple experts (Zhou et al., 2020; Xiang et al., 2020; Wang et al., 2021), or b) applying contrastive learning algorithms (Cui et al., 2021; Zhu et al., 2022) for dealing with long-tailed data.

Although our proposed method is based on the decoupled learning scheme, we would like to clarify that we can combine ours with the existing state-of-the-art frameworks due to its simplicity (i.e., it only needs SWAG posterior and classifier re-training). To this end, we tested ours upon existing state-of-the-art code bases by simply 1) obtaining SWAG posterior from the publicly available pre-trained checkpoint with a few SGD iterations (e.g., 10 training epochs) and 2) re-training the classification layer as we proposed in Section 4.2. We adapted the following implementations:

**Table 8:** Results on CIFAR10-LT and CIFAR100-LT: classification accuracy (ACC), negative log-likelihood (NLL), and expected calibration error (ECE).

| CIFAR10-LT | ACC (↑) Many | Medium | Few | All | NLL (↓) | ECE (↓) |
|---|---|---|---|---|---|---|
| SGD | - | - | - | $72.02_{\pm0.31}$ | $1.443_{\pm0.103}$ | $0.207_{\pm0.005}$ |
| + cRT (Kang et al., 2020) | - | - | - | $80.80_{\pm0.03}$ | $\mathbf{0.598}_{\pm0.017}$ | $\mathbf{0.041}_{\pm0.001}$ |
| + LWS (Kang et al., 2020) | - | - | - | $80.31_{\pm0.07}$ | $\mathbf{0.615}_{\pm0.002}$ | $0.055_{\pm0.002}$ |
| + LA (Menon et al., 2021) | - | - | - | $77.36_{\pm0.14}$ | $0.947_{\pm0.071}$ | $0.146_{\pm0.001}$ |
| + DisAlign (Zhang et al., 2021) | - | - | - | $\mathbf{81.37}_{\pm0.02}$ | $\mathbf{0.600}_{\pm0.025}$ | $0.054_{\pm0.001}$ |
| **SWA (ours)** | - | - | - | $74.86_{\pm0.23}$ | $1.201_{\pm0.083}$ | $0.176_{\pm0.007}$ |
| + cRT (Kang et al., 2020) | - | - | - | $81.72_{\pm0.05}$ | $0.566_{\pm0.006}$ | $0.048_{\pm0.001}$ |
| + LWS (Kang et al., 2020) | - | - | - | $80.53_{\pm0.11}$ | $0.603_{\pm0.005}$ | $0.056_{\pm0.002}$ |
| + LA (Menon et al., 2021) | - | - | - | $81.78_{\pm0.07}$ | $0.733_{\pm0.008}$ | $0.109_{\pm0.000}$ |
| + DisAlign (Zhang et al., 2021) | - | - | - | $81.96_{\pm0.05}$ | $0.574_{\pm0.011}$ | $0.052_{\pm0.000}$ |
| **+ SRepr (ours)** | - | - | - | $\mathbf{82.06}_{\pm0.01}$ | $0.542_{\pm0.010}$ | $\mathbf{0.021}_{\pm0.001}$ |

| CIFAR100-LT | ACC (↑) Many | Medium | Few | All | NLL (↓) | ECE (↓) |
|---|---|---|---|---|---|---|
| SGD | $68.66_{\pm0.38}$ | $38.94_{\pm0.29}$ | $9.50_{\pm0.57}$ | $40.51_{\pm0.32}$ | $3.202_{\pm0.080}$ | $0.333_{\pm0.010}$ |
| + cRT (Kang et al., 2020) | $66.05_{\pm0.05}$ | $42.14_{\pm0.15}$ | $15.76_{\pm0.22}$ | $42.59_{\pm0.13}$ | $3.271_{\pm0.010}$ | $0.317_{\pm0.002}$ |
| + LWS (Kang et al., 2020) | $66.24_{\pm0.05}$ | $41.49_{\pm0.15}$ | $16.95_{\pm0.16}$ | $42.79_{\pm0.11}$ | $3.971_{\pm0.004}$ | $0.377_{\pm0.001}$ |
| + LA (Menon et al., 2021) | $60.09_{\pm0.08}$ | $43.34_{\pm0.17}$ | $27.27_{\pm0.19}$ | $\mathbf{44.37}_{\pm0.15}$ | $\mathbf{2.636}_{\pm0.003}$ | $\mathbf{0.231}_{\pm0.001}$ |
| + DisAlign (Zhang et al., 2021) | $66.90_{\pm0.12}$ | $42.51_{\pm0.12}$ | $16.94_{\pm0.19}$ | $43.37_{\pm0.11}$ | $4.685_{\pm0.004}$ | $0.413_{\pm0.003}$ |
| **SWA (ours)** | $72.36_{\pm0.19}$ | $41.51_{\pm0.21}$ | $7.97_{\pm0.71}$ | $42.34_{\pm0.18}$ | $2.924_{\pm0.030}$ | $0.301_{\pm0.007}$ |
| + cRT (Kang et al., 2020) | $67.37_{\pm0.20}$ | $46.81_{\pm0.35}$ | $20.11_{\pm0.33}$ | $46.00_{\pm0.09}$ | $2.953_{\pm0.004}$ | $0.282_{\pm0.000}$ |
| + LWS (Kang et al., 2020) | $67.09_{\pm0.07}$ | $48.10_{\pm0.07}$ | $23.33_{\pm0.15}$ | $47.31_{\pm0.04}$ | $3.404_{\pm0.003}$ | $0.327_{\pm0.001}$ |
| + LA (Menon et al., 2021) | $62.05_{\pm0.11}$ | $47.11_{\pm0.13}$ | $31.43_{\pm0.24}$ | $47.63_{\pm0.07}$ | $2.224_{\pm0.020}$ | $0.178_{\pm0.002}$ |
| + DisAlign (Zhang et al., 2021) | $67.68_{\pm0.22}$ | $48.36_{\pm0.02}$ | $23.17_{\pm0.40}$ | $47.56_{\pm0.05}$ | $3.889_{\pm0.000}$ | $0.358_{\pm0.000}$ |
| **+ SRepr (ours)** | $66.69_{\pm0.01}$ | $49.91_{\pm0.01}$ | $23.31_{\pm0.11}$ | $\mathbf{47.81}_{\pm0.02}$ | $\mathbf{2.148}_{\pm0.009}$ | $\mathbf{0.149}_{\pm0.002}$ |

**Table 9:** Results for combining ours with the existing state-of-the-art methods on CIFAR100-LT and ImageNet-LT: classification accuracy (ACC), negative log-likelihood (NLL), and expected calibration error (ECE).

| Method | CIFAR100-LT ACC | NLL | ECE | ImageNet-LT ACC | NLL | ECE |
|---|---|---|---|---|---|---|
| RIDE (Wang et al., 2021) | 48.3 | 2.183 | 0.168 | 53.9 | 2.487 | **0.196** |
| + SWA + SRepr (ours) | **49.2** (+0.9) | **2.176** (-0.007) | **0.159** (-0.009) | **54.8** (+0.9) | **2.471** (-0.016) | 0.198 (+0.002) |
| RIDE + CMO (Park et al., 2022) | 48.5 | 2.161 | 0.155 | 54.1 | 2.425 | **0.187** |
| + SWA + SRepr (ours) | **49.1** (+0.6) | **2.051** (-0.110) | **0.118** (-0.037) | **55.0** (+0.9) | **2.421** (-0.004) | 0.189 (+0.002) |
| SADE (Zhang et al., 2022) | 49.9 | 2.015 | 0.120 | 59.0 | 1.826 | 0.060 |
| + SWA + SRepr (ours) | **50.1** (+0.2) | **2.008** (-0.007) | **0.117** (-0.003) | **59.2** (+0.2) | **1.824** (-0.002) | **0.059** (-0.001) |
| BCL (Zhu et al., 2022) | 52.0 | 1.941 | 0.192 | 57.2 | 1.871 | 0.062 |
| + SWA + SRepr (ours) | **52.4** (+0.4) | **1.899** (-0.042) | **0.102** (-0.090) | **57.5** (+0.3) | **1.857** (-0.014) | **0.036** (-0.026) |

- https://github.com/frank-xwang/RIDE-LongTailRecognition

- https://github.com/Vanint/SADE-AgnosticLT

- https://github.com/FlamieZhu/Balanced-Contrastive-Learning

Table 9 demonstrates that ours indeed improves the existing state-of-the-art methods. A marginal improvement upon multi-expert models (i.e., RIDE, RIDE + CMO, and SADE) is probably due to the ensemble model already providing well-calibrated predictions (Lakshminarayanan et al., 2017; Ovadia et al., 2019). Even if SWA-Gaussian well captures a single modality, naive ensembling significantly benefits from the multi-modal Bayesian marginalization (Wilson and Izmailov, 2020). Nevertheless, we believe the clear improvements upon the contrastive learning approach (i.e., BCL), which uses only a single model, bear out the value of the proposed method.

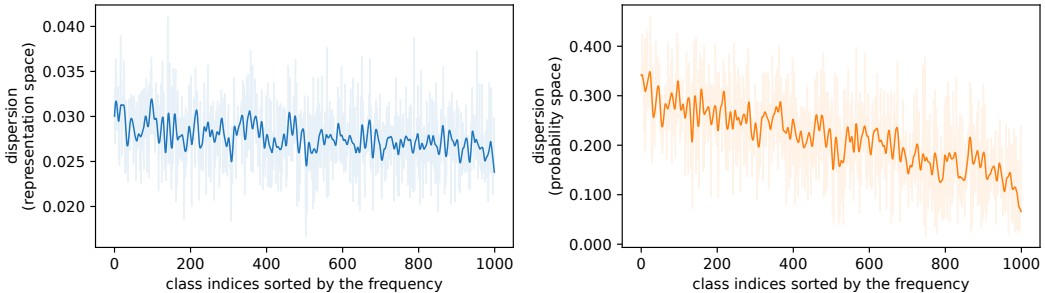

**Figure 5:** The per-class dispersion along with class indices in the representation space (left) and the probability space (right). The results are with ResNet-50 on ImageNet-LT.

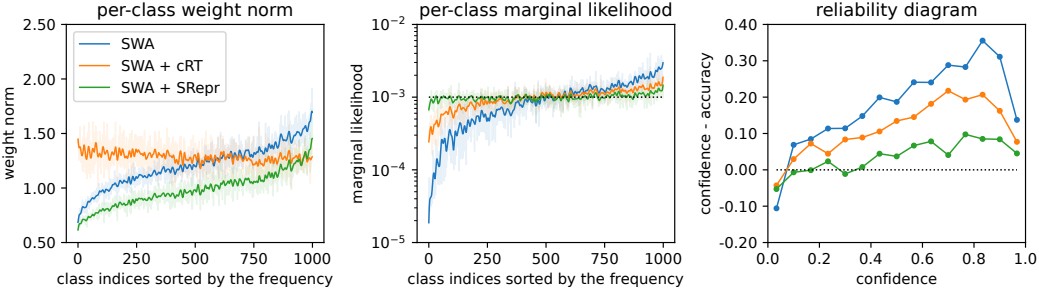

**Figure 6:** The per-class weight norm of the classifier (left), the per-class marginal likelihood of the test predictions (middle), and the reliability diagram for the test predictions (right). The results are with ResNet-50 on ImageNet-LT.

## B.5 FURTHER ANALYSIS ON PROPOSED METHODS

**Measuring dispersion along with class indices**  While we already confirmed a positive correlation between *per-instance* NLL and dispersion in Section 3.2, it would be worth further investigating it in the context of long-tailed recognition. To this end, Fig. 5 depicts the *per-class* dispersion along with class indices sorted by the number of training examples for each class. It clearly shows that the head class tends to have smaller dispersion (especially in the probability space dispersion), which indicates that our method is suitable for long-tailed recognition.

**Per-class weight norms and marginal likelihoods after classifier re-training**  Following previous works (Kang et al., 2020; Ren et al., 2020; Alshammari et al., 2022), we further investigate how our proposed approach affects 1) the per-class weight norm, i.e., $\|\boldsymbol{w}_y\|$ for $y = 1, ..., K$, and 2) the per-class marginal likelihood, i.e., $p(y) = \mathbb{E}_{(\boldsymbol{x}, \cdot) \sim \mathcal{D}_{\text{test}}}[\boldsymbol{p}_{\boldsymbol{\theta}, \boldsymbol{\phi}}^{(y)}(\boldsymbol{x})]$ for $y = 1, ..., K$, where $\mathcal{D}_{\text{test}}$ is a balanced test split. Fig. 6 depicts the per-class weight norm (left) and the per-class marginal likelihood (middle), along with class indices in x-axes sorted by the number of training examples for each class. We also plot the reliability diagram (right; Guo et al., 2017), showing whether the classification model produces well-calibrated predictions.

Alshammari et al. (2022) argued that balanced weight norms of the classifier give tail classes a chance to compete with head classes and produce the ideal marginal likelihood following a uniform distribution. However, Fig. 6 demonstrates that *our re-training method (i.e., SRepr) achieves both the uniform marginal likelihood (middle) and the well-calibrated prediction (right), even though it does not balance weight norms (left).* This phenomenon suggests that our proposed approach works distinctly from the existing works balancing the weight norms for dealing with long-tailed data.

**Training costs compared with vanilla decoupled training.**  Our approach (i.e., SRepr) requires $M$ forward passes of the backbone network during the classifier re-training stage. However, the additional training cost due to these multiple forward passes is not a huge bottleneck since we only

**Table 10:** Training costs compared with vanilla decoupled training.

| Method | Epoch | Wall-clock time | ACC | NLL | ECE |
|---|---|---|---|---|---|
| SWA + cRT | 20 | 647.2 sec | 52.28 | 2.309 | 0.132 |
| **SWA + SRepr (ours)** | 10 | **580.9 sec** | **53.76** | **2.072** | **0.059** |
| SWA + cRT | 40 | 1294.4 sec | 52.47 | 2.284 | 0.124 |
| **SWA + SRepr (ours)** | 20 | **1161.8 sec** | **53.81** | **2.069** | **0.052** |

**Table 11:** Ablation study on alternative way to generate stochastic representation. $\mathcal{F}_m(\boldsymbol{x})$ denotes the $m^{\text{th}}$ stochastic representation for a given input $\boldsymbol{x}$.

| Method | How to generate the stochastic representation? | ACC | NLL | ECE |
|---|---|---|---|---|
| SWA + cRT | - | 52.28 | 2.309 | 0.132 |
| **SWA + SRepr (ours)** | changing parameters, i.e., $\mathcal{F}_m(\boldsymbol{x}) = \mathcal{F}_{\boldsymbol{\theta}_m}(\boldsymbol{x})$, where $\boldsymbol{\theta}_1, ..., \boldsymbol{\theta}_M \stackrel{\text{i.i.d.}}{\sim} q(\boldsymbol{\theta})$. | **53.81** | **2.069** | **0.052** |
| **SWA + SRepr (ours)** | changing inputs, i.e., $\mathcal{F}_m(\boldsymbol{x}) = \mathcal{F}_{\boldsymbol{\theta}_{\text{SWA}}}(\boldsymbol{x}_m)$, where $\boldsymbol{x}_m$ is the $m^{\text{th}}$ augmented version of $\boldsymbol{x}$. | 53.35 | 2.115 | 0.068 |

backpropagate through the last classifier layer while holding the backbone network frozen. To be concrete, Table 10 presents the wall-clock time of re-training classifier with and without SRepr for ResNet-50 on ImageNet-LT when $M$ is 10. While SRepr requires x1.8 times compared to the vanilla classifier re-training (i.e., 32.36 sec/epoch vs. 58.09 sec/epoch), the performance gain achieved by SRepr outweighs the additional training time. That is, even if we run cRT for twice more training epochs than SRepr, it is worse than the performance that SRepr could achieve with the half number of training epochs. We also note that cRT cannot reach the numbers obtained by SRepr even if it is ran for longer training epochs.

**Generating stochastic representation by random augmentation.** We can also generate stochastic representations by changing inputs (i.e., random augmentation) instead of model parameters (i.e., SWA-Gaussian, as we introduced in Eq. (9)). Table 11 shows that the proposed SRepr also works with random augmentation (i.e., random crop augmentation), but it is worse than one we originally proposed. One notable advantage of generating stochastic representations using SWAG is that we do not need to design data-dependent augmentation strategies. This helps when we are to apply our method to the domain for which no straightforward data augmentation strategies are available (e.g., text, graph, and speech data).

**Ablation study on mixup augmentation and our approach.** We also present the ablation study on the mixup augmentation and our approach (i.e., SRepr) since both methods improve the calibration of the classification model. Table 12 clarifies our effectiveness distinct from the mixup augmentation for ResNet-50 on ImageNet-LT; (1) The performance gain is most significant when we use them together. (2) While both the mixup and SRepr improve all metrics upon the baseline, the contribution from SRepr is more significant than that from the mixup.

## C EXPERIMENTAL DETAILS

Code is available at https://github.com/cs-giung/long-tailed-srepr. Our implementations are built on JAX (Bradbury et al., 2018), Flax (Heek et al., 2020), and Optax (Hessel et al., 2020). These libraries are available under the Apache-2.0 license[2]. For ImageNet-LT and iNaturalist-2018, we conduct all experiments on 8 TPUv3 cores, supported by TPU Research Cloud[3].

### C.1 DATASETS

**ImageNet-LT.** It is available at https://github.com/zhmiao/OpenLongTailRecognition-OLTR (Liu et al., 2019). It consists of 115,846 train examples, 20,000 validation examples and 50,000 test

---

[2] https://www.apache.org/licenses/LICENSE-2.0
[3] https://sites.research.google/trc/about/

**Table 12:** Ablation study on mixup augmentation and our approach.

| mixup | **SRepr (ours)** | ACC | NLL | ECE |
|:---:|:---:|---|---|---|
| | | 52.28 | 2.309 | 0.132 |
| ✓ | | 53.64 (+1.36) | 2.131 (-0.178) | 0.079 (-0.053) |
| | ✓ | 53.81 (+1.53) | 2.069 (-0.240) | 0.052 (-0.080) |
| ✓ | ✓ | **54.43** (+2.15) | **1.990** (-0.319) | **0.022** (-0.110) |

examples from 1,000 classes. While the train split is imbalanced, with maximally 1,280 images per class and minimally 5 images per class, the validation and test splits are balanced.

We follow the standard data augmentation policy which consists of random resized cropping with an images size of $224 \times 224 \times 3$ and random horizontal flipping. All images are standardized by subtracting the per-channel mean and dividing the result by the per-channel standard deviation. For per-channel mean and standard deviation values, we stay consistent with the values from the full ImageNet-1k dataset (Russakovsky et al., 2015), i.e., mean of $(0.485, 0.456, 0.406)$ and standard deviation of $(0.229, 0.224, 0.225)$ in RGB order.

**iNaturalist-2018.** It is available at https://github.com/visipedia/inat_comp (Van Horn et al., 2018). It consists of 437,513 train examples, 24,426 validation examples and 149,394 test examples from 8,142 classes. Since the ground-truth labels of the test split are not publicly available, we instead use the balanced validation split as the test split. We apply the same data augmentation for training as that of ImageNet-LT.

**CIFAR10/100-LT.** It is available at https://github.com/kaidic/LDAM-DRW (Cao et al., 2019). It consists of 10,847 train examples and 10,000 test examples from 10/100 classes when an exponential decay with an imbalance factor of $0.01$ is applied. We use a simple data augmentation policy which consists of random padded cropping and random horizontal flipping (He et al., 2016). All images are standardized with the mean of $(0.4914, 0.4822, 0.4465)$ and standard deviation of $(0.2023, 0.1994, 0.2010)$ in RGB order.

## C.2    OPTIMIZATION

**ImageNet-LT and iNaturalist-2018.** Throughout the main experiments on ImageNet-LT and iNaturalist-2018, we use an SGD optimizer with batch size 256, Nesterov momentum 0.9, and a single-cycle cosine decaying learning rate starting from the base learning rate of $0.1$. Unless specified, the optimization for the representation learning stage terminates after 100 training epochs for ImageNet-LT and 200 training epochs for iNaturalist-2018. For the classifier re-training, we introduce an additional 10% training epochs to re-train the classifier.

**CIFAR10/100-LT.** Throughout the additional experiments on CIFAR10/100-LT in Appendix B.2, we apply the same optimization strategy as that of ImageNet-LT and iNaturalist-2018, except for the batch size of $128$ and the baseline learning rate of $0.5$.

## C.3    WEIGHT DECAY (WD)

Weight Decay (WD; Krogh and Hertz, 1991) is the standard regularization technique for training deep neural networks. For instance, we additionally introduce the WD term of $\lambda_{\mathrm{wd}} \|\boldsymbol{\Theta}\|_2^2$ in Eq. (2), where $\lambda_{\mathrm{wd}} > 0$ is a hyperparameter to control the impact of the WD term. Table 13 shows validation accuracy of SGD swept over $\lambda_{\mathrm{wd}} \in \{0.0001, 0.0002, 0.0003, 0.0004, 0.0005\}$. We empirically found that tuning weight decay exerts a strong influence on long-tailed classification performance, as Alshammari et al. (2022) reported. Throughout the paper, we apply $\lambda_{\mathrm{wd}} = 0.0003$ for ImageNet-LT, $\lambda_{\mathrm{wd}} = 0.0001$ for iNaturalist-2018, and $\lambda_{\mathrm{wd}} = 0.0005$ for CIFAR10/100-LT.

**Table 13:** Validation accuracy of SGD with varying weight decay coefficients (i.e., $\lambda_{\mathrm{wd}}$ in Appendix C.3) on ImageNet-LT and iNaturalist-2018.

| | ImageNet-LT | | | | iNaturalist-2018 | | | |
|---|---|---|---|---|---|---|---|---|
| | Many | Medium | Few | All | Many | Medium | Few | All |
| $\lambda_{\mathrm{wd}} = 0.0001$ | 64.72 | 39.27 | 14.66 | 45.73 | $76.31_{\pm0.52}$ | $67.89_{\pm0.18}$ | $62.24_{\pm0.17}$ | $\mathbf{66.52}_{\pm0.05}$ |
| $\lambda_{\mathrm{wd}} = 0.0002$ | 67.99 | 41.38 | 13.93 | 47.90 | $77.48_{\pm0.39}$ | $68.53_{\pm0.14}$ | $61.07_{\pm0.38}$ | $66.50_{\pm0.17}$ |
| $\lambda_{\mathrm{wd}} = 0.0003$ | $69.08_{\pm0.36}$ | $41.66_{\pm0.35}$ | $12.67_{\pm0.52}$ | $\mathbf{48.28}_{\pm0.22}$ | 77.91 | 67.37 | 58.46 | 64.93 |
| $\lambda_{\mathrm{wd}} = 0.0004$ | 69.29 | 41.16 | 10.50 | 47.83 | 77.16 | 65.54 | 54.37 | 62.32 |
| $\lambda_{\mathrm{wd}} = 0.0005$ | 69.44 | 40.24 | 8.93 | 47.23 | 77.20 | 63.70 | 50.85 | 60.01 |

**Table 14:** Validation accuracy of SWA with varying SWA learning rates (i.e., $\eta_{\mathrm{SWA}}$ in Appendix C.4) on ImageNet-LT and iNaturalist-2018.

| | ImageNet-LT | | | | iNaturalist-2018 | | | |
|---|---|---|---|---|---|---|---|---|
| | Many | Medium | Few | All | Many | Medium | Few | All |
| $\eta_{\mathrm{SWA}} = 0.020$ | 70.16 | 40.69 | 10.50 | 47.94 | - | - | - | - |
| $\eta_{\mathrm{SWA}} = 0.010$ | $70.02_{\pm0.49}$ | $41.63_{\pm0.34}$ | $11.55_{\pm0.28}$ | $\mathbf{48.47}_{\pm0.22}$ | 77.47 | 67.69 | 61.49 | 66.25 |
| $\eta_{\mathrm{SWA}} = 0.005$ | 69.62 | 41.55 | 12.03 | 48.35 | $77.26_{\pm0.25}$ | $68.23_{\pm0.25}$ | $61.87_{\pm0.13}$ | $\mathbf{66.65}_{\pm0.10}$ |
| $\eta_{\mathrm{SWA}} = 0.001$ | - | - | - | - | 76.60 | 67.86 | 61.71 | 66.33 |

## C.4 STOCHASTIC WEIGHT AVERAGING (SWA)

Stochastic Weight Averaging (SWA; Izmailov et al., 2018) has three hyper-parameters; 1) when does the averaging phase start? 2) how frequently update the moving average? and 3) how to set the learning rate during the averaging phase? In response, we follow the instruction from (Izmailov et al., 2018); 1) the averaging phase starts at 75% of the training epoch, 2) we capture parameters at each epoch for averaging, and 3) we use the high constant learning rate $\eta_{\mathrm{SWA}}$ during the averaging phase. Table 14 shows validation accuracy of SWA swept over $\eta_{\mathrm{SWA}} \in \{0.001, 0.005, 0.010, 0.020\}$ on ImageNet-LT. Throughout the paper, we use $\eta_{\mathrm{SWA}} = 0.010$ for ImageNet-LT, $\eta_{\mathrm{SWA}} = 0.005$ for iNaturalist-2018, and $\eta_{\mathrm{SWA}} = 0.1$ for CIFAR10/100-LT.

---

**Algorithm 1** Decoupled training w/ SWA + SRepr (ours).

---

**Require:** Train dataset $\mathcal{D} = \{(\boldsymbol{x}_i, y_i)\}_{i=1}^N$, the feature extractor $\mathcal{F}$ parameterized by $\boldsymbol{\theta}$, and the linear classification layer parameterized by $\boldsymbol{\phi} = (\boldsymbol{w}_k, b_k)_{k=1}^K$.

**Require:** The number of representation learning steps $T_1$, the number of classifier retraining steps $T_2$, the learning rate schedule $\eta_t$, moment update start time $T_{\text{SWA}}$ and update frequency $f_{\text{SWA}}$.

**Ensure:** Parameters $(\boldsymbol{\theta}^*, \boldsymbol{\phi}^{**})$, trained in decoupled learning scheme.

1: **// Representation learning stage**
2: Initialize the parameters $\boldsymbol{\Theta}_0 = (\boldsymbol{\theta}_0, \boldsymbol{\phi}_0)$ to random values.
3: Initialize the first and second moments $\boldsymbol{\Theta}_{\text{SWA}} = \boldsymbol{0}$, $\boldsymbol{\Theta}'_{\text{SWA}} = \boldsymbol{0}$, and $n_{\text{SWA}} = 0$.
4: **for** $t \in \{1, ..., T_1\}$ **do**
5:     Sample mini-batch $\mathcal{B} \subset \mathcal{D}$.
6:     Update the parameters with stochastic gradients,

$$\boldsymbol{\Theta}_t \leftarrow \boldsymbol{\Theta}_{t-1} - \eta_t \nabla_{\boldsymbol{\Theta}} \mathcal{L}(\boldsymbol{\Theta})\big|_{\boldsymbol{\Theta}=\boldsymbol{\Theta}_{t-1}}, \tag{34}$$

    where the loss is defined as

$$\mathcal{L}(\boldsymbol{\Theta}) = \mathbb{E}_{(\boldsymbol{x},y)\sim\mathcal{B}} \left[ -\log \boldsymbol{p}^{(y)}(\boldsymbol{x}; \boldsymbol{\Theta}) \right]. \tag{35}$$

7:     **if** $t > T_{\text{SWA}}$ and $\text{MOD}(t, f) = 0$ **then**
8:         Update the first and second moments via moving average,

$$\boldsymbol{\Theta}_{\text{SWA}} \leftarrow (n_{\text{SWA}}\boldsymbol{\Theta}_{\text{SWA}} + \boldsymbol{\Theta}_t)/(n_{\text{SWA}} + 1), \tag{36}$$

$$\boldsymbol{\Theta}'_{\text{SWA}} \leftarrow (n_{\text{SWA}}\boldsymbol{\Theta}'_{\text{SWA}} + \boldsymbol{\Theta}_t^2)/(n_{\text{SWA}} + 1), \tag{37}$$

$$n_{\text{SWA}} \leftarrow n_{\text{SWA}} + 1. \tag{38}$$

9:     **end if**
10: **end for**

11: **// Classifier retraining stage**
12: We have the pre-trained parameters $\boldsymbol{\Theta}_{T_1}$, where $\boldsymbol{\Theta}_{T_1} \leftarrow \boldsymbol{\Theta}_{\text{SWA}}$ if we applied SWA.
13: We also have the approximate posterior over the feature extractor parameters, i.e.,

$$q(\boldsymbol{\theta}|\mathcal{D}) = \mathcal{N}(\boldsymbol{\theta}|\boldsymbol{\theta}_{\text{SWA}}, \boldsymbol{\Sigma}_{\text{SWAG}}), \text{ where } \boldsymbol{\Sigma}_{\text{SWAG}} = \text{diag}(\boldsymbol{\theta}'_{\text{SWA}} - \boldsymbol{\theta}_{\text{SWA}}^2). \tag{39}$$

14: **for** $t \in \{T_1 + 1, ..., T_1 + T_2\}$ **do**
15:     Sample mini-batch $\mathcal{B} \subset \mathcal{D}$.
16:     Update the parameters with stochastic gradients (with some balancing strategy),

$$\boldsymbol{\phi}_t \leftarrow \boldsymbol{\phi}_{t-1} - \eta_t \nabla_{\boldsymbol{\phi}} \mathcal{L}(\boldsymbol{\phi})\big|_{\boldsymbol{\phi}=\boldsymbol{\phi}_{t-1}}, \tag{40}$$

    where the loss is defined as

$$\mathcal{L}(\boldsymbol{\phi}) = \mathbb{E}_{(\boldsymbol{x},y)\sim\mathcal{B}} \left[ -\log \boldsymbol{p}^{(y)}(\boldsymbol{x}; (\boldsymbol{\theta}^*, \boldsymbol{\phi})) \right], \tag{41}$$

$$\text{or } \mathcal{L}(\boldsymbol{\phi}) = \mathbb{E}_{(\boldsymbol{x},y)\sim\mathcal{B}} \left[ 0.5\hat{\mathcal{L}}_{\text{CE}} + 0.5\hat{\mathcal{L}}_{\text{KD}} \right], \text{ if we applied SRepr.} \tag{42}$$

    Here, the first term is

$$\hat{\mathcal{L}}_{\text{CE}} = \mathbb{E}_{(\boldsymbol{x},y)\sim q(\boldsymbol{\theta}|\mathcal{D})} \left[ -\log \boldsymbol{p}^{(y)}(\boldsymbol{x}; (\boldsymbol{\theta}, \boldsymbol{\phi})) \right]. \tag{43}$$

    Refer to Eq. (16) for the detailed definition of the second term.
17: **end for**

18: **Return** $\boldsymbol{\theta}^* = \boldsymbol{\theta}_{T_1}$ and $\boldsymbol{\phi}^{**} = \boldsymbol{\phi}_{T_1+T_2}$, trained in decoupled learning scheme.

---

