# OpenReview forum: "Decoupled Training for Long-Tailed  Classification With Stochastic Representations"
_ICLR.cc/2023/Conference — ICLR 2023 poster_

### Official Review · Reviewer_eyhi · 2022-10-18

**Confidence:** 4
**Clarity, Quality, Novelty And Reproducibility:** This paper is clearly written, code i…
**Correctness:** 3
**Technical Novelty And Significance:** 3
**Empirical Novelty And Significance:** 3
**Recommendation:** 5

**Strength And Weaknesses:**

I think this paper is very well-written and well-organized. The preliminary experiments on the effect of stochastic representation (Stochastic Weight Average, SWA) on decoupled long-tail learning methods are detailed and intuitive. They start with testing directly applying stochastic representations to single stage and two stage training and acquired results that promote further development of SWA on two-stage/decouple methods. The idea of using Gussian SWA (SWAG) as a measurement of prediction certainty, although not new, but directly connected to the intuition that tail class predictions are usually less certain. A robust certainty measurement is always a challenge in long-tail recognition, and I think using SWAG is an interesting proposition. The discussion and explanation of the limitations of direct usage of SWAG are clear, because it needs multiple runs to get a stochastic distribution. And this leads to the proposition of the proposed self-distillation method, which has multiple teachers during training for a SWAG distribution, and has a student, that learns from the stochastic teachers. Again, this multi-teacher single student framework is not new (e.g., mean teachers in semi-supervised learning). But it aligns well with the challenges mentioned in direct usage of SWAG. The experiments are extensive. Even the state-of-the-art multi-expert methods like  RIDE are compared. However, the improvements, especially the improvements over multi-expert methods, are limited (<1% on average). The biggest drawback of this method is that it is relatively too complicated (multiple teach networks need to be trained for SWAG to work), and, thus, not compatible with existing multi-expert models, which leads to limited improvements. The proposed multi-teacher single student framework itself is already a multi-expertish model.

**Summary Of The Paper:**

This paper studies the effect of stochastic representations in long-tail recognition scenarios and designs a self-distillation method to adapt stochastic representations to long-tail learning.

**Summary Of The Review:**

This paper overall is a good contribution to the community for its detailed experiments on the effect of stochastic representations on long-tailed learning, and the idea of using stochastic representations as a measurement of confidence. However, the proposed method is relatively complicated and yields limited improvements over state-of-the-art methods.

---

> ### Author Response · Authors · 2022-11-15
> **Response to eyhi**
>
> Thank you for your supportive and constructive comments. We hope our general response and the below replies further alleviate your concerns.
>
>
> > However, the improvements, especially the improvements over multi-expert methods, are limited (<1% on average). The biggest drawback of this method is that it is relatively too complicated (multiple teach networks need to be trained for SWAG to work), and, thus, not compatible with existing multi-expert models, which leads to limited improvements. The proposed multi-teacher single student framework itself is already a multi-expertish model.
>
> Again, training SWA-Gaussian has virtually no additional cost over conventional training (except for a memory cost required for storing moving averages of parameters). It is a clear advantage distinct from existing practices for knowledge distillation (e.g., Wang et al. (2021) present a 3-expert student model distilled from an independently trained 6-experts teacher model, which requires a massive computation cost). We believe our method is neither complicated nor computationally expensive, as summarized in Algorithm 1 (along with the first paragraph of Appendix B.5).
>
> Regarding the marginal improvements upon multi-expert models, we supplemented Appendix B.4. To summarize, that is probably due to the ensemble model already providing well-calibrated predictions (Lakshminarayanan et al., 2017; Ovadia et al., 2019). Even if SWA-Gaussian well captures a single modality, naive ensembling significantly benefits from the multi-modal Bayesian marginalization (Wilson and Izmailov, 2020). Nevertheless, we believe the clear improvements upon the single model (i.e., BCL) make the proposed approach valuable.
>
>
> ----------
>
> References
>
> - (Wang et al., 2021) Long-tailed Recognition by Routing Diverse Distribution-Aware Experts.
> - (Lakshminarayanan et al., 2017) Simple and Scalable Predictive Uncertainty Estimation using Deep Ensembles.
> - (Ovadia et al., 2019) Can You Trust Your Model's Uncertainty? Evaluating Predictive Uncertainty Under Dataset Shift.
> - (Wilson and Izmailov, 2020) Bayesian Deep Learning and a Probabilistic Perspective of Generalization.

---

> > ### Comment · Reviewer_eyhi · 2022-11-30
> > **Response**
> >
> > Thanks for the rebuttal. First, I think the increased memory cost itself already proves the complexity of the method. It doesn't necessarily need to be slow, it is more like the difference between single gpu vs multi gpus. Second, I think because this method has multiple teachers, so it is not fair to compare single models, and the marginal improvement over existing multi-expert models is there. After reading other reviewers' comments, I changed my rate to 5.

---

> > > ### Author Response · Authors · 2022-12-12
> > > **Response to eyhi**
> > >
> > > > Thanks for the rebuttal. First, I think the increased memory cost itself already proves the complexity of the method. It doesn't necessarily need to be slow, it is more like the difference between single gpu vs multi gpus.
> > >
> > > It would be appreciated if you further clarify what "the increased memory cost" means. Again, SWAG-Diagonal only requires storing two additional copies of model parameters (for the first and the second moments), which do not have to be loaded on GPU since we update them via the running average (and thus, the number of GPUs is irrelevant). The following quote from Maddox et al. (2019) clarifies this point; “Constructing the SWAG-Diagonal posterior approximation requires storing two additional copies of DNN weights. Note that these models do not have to be stored on the GPU. The additional computational complexity of constructing SWAG-Diagonal compared to standard training is negligible, as it only requires updating the running averages of weights once per epoch.”
> > >
> > > > Second, I think because this method has multiple teachers, so it is not fair to compare single models, and the marginal improvement over existing multi-expert models is there. After reading other reviewers' comments, I changed my rate to 5.
> > >
> > > Although multiple settings of model parameters (what you call "multiple teachers") get involved in classifier re-training, we get a single solution at the end. If ours requires far more costs to obtain such a single solution, it would be unfair to compare ours to the baseline approaches (e.g., CRT). However, 1) SWAG-Diag has virtually no additional cost over conventional training with SGD, and 2) Appendix B.5 clarifies that the proposed SRepr only requires the same level of costs as the baseline classifier re-training method. Thus it is reasonable to compare ours to the baseline approaches. Note that this is not the case for the existing works requiring independently trained multiple teachers for distillation (Iscen et al., 2021; Wang et al., 2021).

---

### Official Review · Reviewer_DCKB · 2022-10-22

**Confidence:** 5
**Clarity, Quality, Novelty And Reproducibility:** The code is provided in the supplemen…
**Correctness:** 3
**Technical Novelty And Significance:** 3
**Empirical Novelty And Significance:** 3
**Recommendation:** 5

**Details Of Ethics Concerns:**

Some discussion is added.

**Strength And Weaknesses:**

Strength

(1) The paper writes clear and is easy to follow.

(2) The method should be general and is expected to work well with previous methods.

Weakness

(1) This paper use SWA to enhance the representation ability of deep models and a distillation method is proposed to reduce predictive variance. However, comparisons with most related work are missed.
      I. The state-of-the-art methods, like PaCo [1].
      II. distillation-based methods, like CBD [2].


[1] Parametric Contrastive Learning. ICCV 2021.
[2] Class-Balanced Distillation for Long-Tailed Visual Recognition. BMVC 2021.

(2) About the distillation objective.  Why do the teacher prediction probabilities obey the Dirichlet distribution?

(3) As shown in Figure2, as the number of stochastic representations increases, the changes of ECE and NLL are limited.

**Summary Of The Paper:**

This paper aims to solve the fundamental problem --- long-tailed recognition. It analyzes SWA/SWAG with long-tailed data.

（1) Classifier re-training can make better use of SWA for long-tailed data.
  (2) The positive correlation between NLL and dispersion is observed. Based on this phenomenon, a self-distillation strategy is developed to reduce predictive variance.
  (3) Experiments on ImageNet-LT and iNaturalist 2018 show some improvements.


**Summary Of The Review:**

The paper investigates SWA/SWAG for long-tailed data. However, some important comparisons are missed.
If the concerns are addressed, I'm very glad to raise my score.

---

> ### Author Response · Authors · 2022-11-15
> **Response to DCKB**
>
>
> > This paper use SWA to enhance the representation ability of deep models and a distillation method is proposed to reduce predictive variance. However, comparisons with most related work are missed; I. The state-of-the-art methods, like PaCo (Parametric Contrastive Learning. ICCV 2021); II. distillation-based methods, like CBD (Class-Balanced Distillation for Long-Tailed Visual Recognition. BMVC 2021).
>
> As we mentioned in the general response, Appendix B.4 now clarifies that we could apply the proposed method to the existing state-of-the-art frameworks. We believe that the results showing a synergy between ours and BCL (Zhu et al., 2022), a more up-to-date contrastive-learning-based approach compared to PaCo (Cui et al., 2021), resolve the first issue you mentioned. Regarding CBD (Iscen et al., 2021), we would like to clarify that there is a clear difference between ours and theirs in that our method does not require a costly teacher model. The revised version of our paper now clarifies this in Section 5 (i.e., related works). Thank you for your constructive comments that make our paper solid.
>
>
> > About the distillation objective. Why do the teacher prediction probabilities obey the Dirichlet distribution?
>
> It is natural to assume the teacher and student outputs as Dirichlet random vectors since they are categorical probability vectors, and this is a common setting in various works dealing with Bayesian neural networks with priors placed on output probabilities (Malinin and Gales, 2018; Malinin et al., 2020; Ryabinin et al., 2021).
>
>
> > As shown in Figure2, as the number of stochastic representations increases, the changes of ECE and NLL are limited.
>
> Figure 3 shows that just ensembling stochastic representations over the existing decision boundary (i.e., Equation 8) could improve the uncertainty estimates. In the context of Bayesian Model Averaging, it is natural that the metrics get saturated as the number of samples for Monte Carlo integration increases. For instance, Wilson and Izmailov (2020) include a lot of plots showing the convergence of the predictive distributions for Bayesian neural networks.
>
>
> ----------
>
> References
>
> - (Zhu et al., 2022) Balanced Contrastive Learning for Long-Tailed Visual Recognition.
> - (Cui et al., 2021) Parametric Contrastive Learning.
> - (Iscen et al., 2021) Class-Balanced Distillation for Long-Tailed Visual Recognition.
> - (Malinin and Gales, 2018) Predictive Uncertainty Estimation via Prior Networks.
> - (Malinin et al., 2020) Ensemble Distribution Distillation.
> - (Ryabinin et al., 2021) Scaling Ensemble Distribution Distillation to Many Classes with Proxy Targets.

---

> > ### Comment · Reviewer_DCKB · 2022-12-11
> > **Response to the authors**
> >
> > Thanks for the reply from the authors.
> >
> > I have gone through comments from other reviewers. I agree with the Reviewer fqWB and the Reviewer eyhi that improvements obtained from the proposed method are limited, especially when compared with strong baselines, for example, only 0.2% and 0.3% for SADE and BCL. It is hard to conclude that such improvements are from the proposed method rather than random variance.
> >
> > The authors add comparisons with SADE and BCL in the latest version. However, the limited improvements are unconvinced for me.

---

> > > ### Author Response · Authors · 2022-12-12
> > > **Response to DCKB**
> > >
> > >
> > > > Thanks for the reply from the authors.
> > > > I have gone through comments from other reviewers. I agree with the Reviewer fqWB and the Reviewer eyhi that improvements obtained from the proposed method are limited, especially when compared with strong baselines, for example, only 0.2% and 0.3% for SADE and BCL. It is hard to conclude that such improvements are from the proposed method rather than random variance.
> > > > The authors add comparisons with SADE and BCL in the latest version. However, the limited improvements are unconvinced for me.
> > >
> > > The main message of Appendix B.4 is giving promise that our approach can complement other methods, as Reviewer 1FXP commented. Here, we would like to highlight the definite improvement in ECE on BCL, displayed in Table 9; ours improved ECE by 46.9 percent on CIFAR100-LT (0.192 to 0.102) and 41.9 percent on ImageNet-LT (0.062 to 0.036). It demonstrates the proposed re-training strategy is beneficial, especially for calibrating a single deep neural classifier model.
> > >
> > > Also, we stated the limited improvements just as it is when ours complements multi-expert models (e.g., SADE), i.e., the ensemble model already provides well-calibrated predictions (Lakshminarayanan et al., 2017; Ovadia et al., 2019). Nevertheless, this is not the main drawback of the proposed method since it is well worthwhile improving the calibration of the single model (e.g., BCL), and we should not take the proposed method as a panacea for all problems. We believe that pointing out such limitations of the presented and giving some educated guesses contribute to a machine learning community.
> > >
> > > Besides, note that current results in Table 9 have been obtained by simply fine-tuning the existing pre-trained checkpoints. There is still room for improvements in our method (i.e., SWA + SRepr) by running it from scratch; constructing SWAG-Diag from the existing SGD solution is suboptimal.

---

### Official Review · Reviewer_fqWB · 2022-10-24

**Confidence:** 3
**Correctness:** 3
**Technical Novelty And Significance:** 2
**Empirical Novelty And Significance:** 2
**Recommendation:** 5

**Clarity, Quality, Novelty And Reproducibility:**

The clarity, quality, novelty and reproducibility are okay. The novelty is incremental as the paper seems to apply Stochastic Weight Averaging to improve the generalizability of learned features to improve long-tailed recognition performance. However, the performance improvement is marginal especially given the large computation cost. Code is provided but it is unclear whether running it reproduces the reported results. I'd suggest attaching something like jupyter notebook for better demonstration; authors should not expect reviewers to run their code smoothly.

**Details Of Ethics Concerns:**

No ethics issues as I am aware.

**Strength And Weaknesses:**

Strength
- Studying how generalized features improves LTR is interesting.
- Studying uncertainty estimation in the context of LTR is interesting.


Below are some weaknesses.

- While "the success of decoupling naturally motivates obtaining more informative representations from which the classifier re-training can benefit", using SWA seems like less well-motivated. Instead, this motivation should suggest learn more generalized features using more data sources (e.g., external data) and more general loss functions (e.g., contrastive learning). I would like the authors to discuss this further, otherwise the paper just seems incremental especially when it states "to the best of our knowledge, it has never been explored for long-tailed classification problems.", and in the conclusion "to the best of our knowledge, this is the first attempt to introduce SWA into long-tailed learning."

- While the paper is motivated to learn "generalizable" features, a recent work [R1] shows that learning a backbone which has "balanced-norm" filters significantly boosts LTR performance. Moreover, [R1] simply tunes weight decay to regularize backbone's norms and achieves the state-of-the-art. This is an open questions -- whether to learn "balanced backbone" or more "generalized backbone" to achieve better LTR performance? Can authors discuss and show insights?

[R1] Alshammari, et al., "Long-Tailed Recognition via Weight Balancing", CVPR 2022

- While it is interesting to study uncertainty estimation in the context of LTR, the paper does not sufficiently analyze the results. For example, it is unclear how good uncertainty estimation for tail classes vs. head classes. Authors are encouraged to dive into this part and provide insightful discussions.

typos
- Figure 2 caption: "asdfsadf".

**Summary Of The Paper:**

The paper focuses on improving features to be more generalizable to achieve better performance in Long-Tailed Recognition (LTR). In particular, the paper adopts Stochastic Weight Averaging (SWA) for improving the generalization of deep neural networks. Moreover, the paper proposes a classifier re-training algorithm based on the learned representation. The algorithm uses a Gaussian perturbed SWA, and a self-distillation strategy that can harness the diverse stochastic representations based on uncertainty estimates to build more robust classifiers. Experiments on two benchmarks show that the proposed method improves upon previous methods in terms of both prediction accuracy and uncertainty estimation.

**Summary Of The Review:**

The paper focuses on using Stochastic Weight Averaging (SWA) to improve features used for long-tailed recognition (LTR). The motivation is to learn more generalizable features for LTR. SWA is an existing method that learns more general features and the paper claims that "to the best of our knowledge, this is the first attempt to introduce SWA into long-tailed learning". However, SWA seems weak as it achieves only marginal improvement given the large added computation burden. Therefore, the rating of the paper is "5: marginally below the acceptance threshold"

---

> ### Author Response · Authors · 2022-11-15
> **Response to fqWB (1/2)**
>
>
> > While "the success of decoupling naturally motivates obtaining more informative representations from which the classifier re-training can benefit", using SWA seems like less well-motivated. Instead, this motivation should suggest learn more generalized features using more data sources (e.g., external data) and more general loss functions (e.g., contrastive learning). I would like the authors to discuss this further, otherwise the paper just seems incremental especially when it states "to the best of our knowledge, it has never been explored for long-tailed classification problems.", and in the conclusion "to the best of our knowledge, this is the first attempt to introduce SWA into long-tailed learning."
>
> Thank you for your suggestion. We agree that we might take a different approach, such as using external data sources or employing a contrastive learning framework. However, we think that using SWA and SWA-Gaussian is clearly motivated; SWA is arguably one of the most effective yet simple algorithms that can significantly improve the generalization performance of deep neural networks, as proven in many works (Mandt et al., 2017; Izmailov et al., 2018; Cha et al., 2021; Caldarola et al., 2022) without any external data or contrastive learning. This benefit of SWA is significant, especially for imbalanced data, given that assuming the existence of external data or the feasibility of contrastive learning is not always amenable. Also, as we empirically show through the experiments, the suboptimality of classifiers is deeply connected to the uncertainties of the representations, and our work shows that this can effectively be alleviated through the proposed SWA-Gaussian and self-distillation framework. Last but not least, as we verify with the additional experiments in Appendix B.4, ours can be combined with existing algorithms taking different approaches (Wang et al., 2021; Zhang et al., 2022; Zhu et al., 2022).
>
>
> > While the paper is motivated to learn "generalizable" features, a recent work (Alshammari et al., 2022) shows that learning a backbone which has "balanced-norm" filters significantly boosts LTR performance. Moreover, Alshammari et al. (2022) simply tunes weight decay to regularize backbone's norms and achieves the state-of-the-art. This is an open questions -- whether to learn "balanced backbone" or more "generalized backbone" to achieve better LTR performance? Can authors discuss and show insights?
>
> Alshammari et al. (2022) argued that "tuning weight decay is sufficient to learn a generalizable feature representation." Thus, the "balanced-norm" approach you mentioned is another way to construct a generalizable feature extractor, separate from applying SWA as we proposed.
>
> Regarding Alshammari et al. (2022) you mentioned, it would be good practice to provide additional insights into our method in similar respect they discussed. To this end, we empirically examined the following as previous works have done (Kang et al., 2019; Ren et al., 2020; Alshammari et al., 2022):
>
> - According to Kang et al. (2019), the weight norm of head classes tends to be larger than the weight norm of tail classes (and this is also the main experiment in Alshammari et al. (2022)). How does our classifier re-training method affect the weight norms?
> - According to Ren et al. (2020), the ideal marginal likelihood should follow a uniform distribution (and this is also the main experiment in Alshammari et al. (2022)). Does ours achieve this better, in other words, does ours provide uniform marginal likelihood?
>
> We added these experiments in Appendix A.1. To summarize, our re-training method (i.e., SRepr) achieves both the uniform marginal likelihood and the well-calibrated prediction, even though it does not balance weight norms. This phenomenon suggests that our proposed approach works distinctly from the existing works balancing the weight norms for dealing with long-tailed data.
>
> > While it is interesting to study uncertainty estimation in the context of LTR, the paper does not sufficiently analyze the results. For example, it is unclear how good uncertainty estimation for tail classes vs. head classes. Authors are encouraged to dive into this part and provide insightful discussions.
>
> Thank you for your constructive comment. We agree that further investigation in the context of LTR makes the paper solid. Following your suggestion, we added the "measuring dispersion along with class indices" paragraph in Appendix B.4, which demonstrates the head class tends to have smaller dispersion and that our method is thus suitable for long-tailed recognition.

---

> > ### Author Response · Authors · 2022-11-15
> > **Response to fqWB (2/2)**
> >
> >
> > > However, SWA seems weak as it achieves only marginal improvement given the large added computation burden. Therefore, the rating of the paper is "5: marginally below the acceptance threshold”
> >
> > We respectfully disagree with this point; training SWA-Gaussian with a diagonal covariance has virtually no additional cost over conventional training with SGD (Izmailov et al., 2018; Maddox et al., 2019). As presented in Appendix A.9, SWA-Gaussian only requires extra space to store the parameters’ first and second moments. Also, the main contribution of our paper is a classifier re-training algorithm based on stochastic representations obtained from SWA-Gaussian (i.e., SRepr), not SWA-Gaussian itself.
> >
> > Again, additional training costs of our approach compared with vanilla decoupled training come from multiple forward passes of the backbone network during the classifier re-training stage. However, in Appendix A.7, we empirically demonstrated that the additional training cost due to these multiple forward passes is not that big. Consequently, we believe the proposed method is a simple yet effective strategy in decoupled learning scheme for long-tailed learning.
> >
> >
> > ----------
> >
> > References
> >
> > - (Mandt et al., 2017) Stochastic Gradient Descent as Approximate Bayesian Inference.
> > - (Izmailov et al., 2018) Averaging Weights Leads to Wider Optima and Better Generalization.
> > - (Cha et al., 2021) SWAD: Domain Generalization by Seeking Flat Minima.
> > - (Caldarola et al., 2022) Improving Generalization in Federated Learning by Seeking Flat Minima.
> > - (Wang et al., 2021) Long-tailed Recognition by Routing Diverse Distribution-Aware Experts.
> > - (Zhang et al., 2022) Self-Supervised Aggregation of Diverse Experts for Test-Agnostic Long-Tailed Recognition
> > - (Zhu et al., 2022) Balanced Contrastive Learning for Long-Tailed Visual Recognition
> > - (Alshammari et al., 2022) Long-Tailed Recognition via Weight Balancing.
> > - (Kang et al., 2019) Decoupling Representation and Classifier for Long-Tailed Recognition.
> > - (Ren et al., 2020) Balanced Meta-Softmax for Long-Tailed Visual Recognition.

---

### Official Review · Reviewer_1FXP · 2022-10-24

**Confidence:** 5
**Clarity, Quality, Novelty And Reproducibility:** Generally clear. Good quality. Novel …
**Correctness:** 3
**Technical Novelty And Significance:** 3
**Empirical Novelty And Significance:** 3
**Recommendation:** 8

**Strength And Weaknesses:**

trengths:
+ A novel approach to obtaining better representations / decision boundaries for long-tailed visual recognition.
+ Positive results over the baselines.

Weaknesses:

1. The paper identifies the main source of imbalanced learning as the decision boundary. However, the recent view identifies the culprit as the representation learning stage and suggests that features of the under-represented class are somehow sub-optimal, which leads to worse decision boundaries for the under-represented class. I suggest the authors to tone down "the main bottleneck is the decision boundary" claim.

2. Fig 2: It is not clear why the dispersion of models along the training trajectory should be correlated with NLL.

3. The proposed approach is computationally too expensive. One can easily obtain a set of alternative representations through dropout &  perform distillation. Would this not be cheaper? A discussion on complexity is necessary.

4. The experimental evaluation needs improvement.

4.1. The SOTA comparison in Table 4 seems to have unfair aspects: In some cases, less training epochs are used with the compared baseline (e.g. the comparison against KCL) and more are used in some others (Liu et al. (2021)).

4.2. The experimental evaluation is missing important comparisons with more recent approaches from 2022. No approach is compared against from 2022.

4.3. The experimental evaluation is missing comparisons on CIFAR10LT & CIFAR100LT. This is important to get a better feeling about how the method performs in a small-scale problem (CIFAR10LT) and a middle/large-scale problem such as CIFAR100LT. In fact, CIFAR100LT is surprisingly challenging for some problems and you can obtain unexpected results.


Minor comments:
- "Confirming that SWA can benefit to long-tailed classification" => "Confirming that SWA can provide benefit to long-tailed classification".
- Eq 2: "x" is forgotten after the dot.
- "parameter θ" => "parameters θ".
- "parameter ϕ" => "parameters ϕ".
- Eq 4: A minor detail but this should include a step size.
- Eq 6: Θ′ => not introduced.
- Below Eq 9: "predictive uncertainty of x," => Predictive or epistemic? They are not the same. Two lines above Eq 9, you said epistemic.
- Figure 2: "asdfsadf." => Agreed :)
- Fig 3: ERM not introduced.

**Summary Of The Paper:**

In this paper, the authors study the problem of long-tailed visual classification. To be specific, they propose obtaining a stochastic model for representations through stochastic weight averaging (SWA) and retraining a robust classifier by distilling these representations.

**Summary Of The Review:**

Novel approach but there are concerns about the evaluation.

---

> ### Author Response · Authors · 2022-11-15
> **Response to 1FXP (1/2)**
>
>
> > The paper identifies the main source of imbalanced learning as the decision boundary. However, the recent view identifies the culprit as the representation learning stage and suggests that features of the under-represented class are somehow sub-optimal, which leads to worse decision boundaries for the under-represented class. I suggest the authors to tone down "the main bottleneck is the decision boundary" claim.
>
> We would appreciate you introducing us to such papers you mentioned. It seems that the existing works (i) enhance the feature extractor by considering the under-represented classes during the representation learning stage, and thus, (ii) can obtain better decision boundaries.
>
> If so, we believe that our method shares some spirits with them. Specifically, ours also (i) consider the under-represented instances having dispersed stochastic representations after the representation learning stage (with SWA-Gaussian), and thus, (ii) can obtain better decision boundaries by considering the under-represented instances during the classifier learning stage. As a result, both the existing works you mentioned and our work still serve the same purpose; obtaining better decision boundaries.
>
>
> > Fig 2: It is not clear why the dispersion of models along the training trajectory should be correlated with NLL.
>
> The main message of SWA-Gaussian is that we can approximate the posterior of model parameters with a Gaussian distribution using samples from the SGD training trajectory (Mandt et al., 2017; Maddox et al., 2019). Thus, it would be better to note that the models sampled from the approximate posterior distribution (rather than the point on the training trajectory) produce the stochastic representation. As we stated in Section 3.2, our hypothesis on stochastic representations is that they reflect the uncertainty of inputs through dispersion. Indeed, Figure 2 empirically confirms the positive correlation between the per-instance negative log-likelihood (which quantifies the per-instance uncertainty) and the dispersion.
>
>
> > The proposed approach is computationally too expensive. One can easily obtain a set of alternative representations through dropout & perform distillation. Would this not be cheaper? A discussion on complexity is necessary.
>
> As you suggested, one might use MC dropout (Gal and Ghahramani, 2016) instead of SWA-Gaussian, but that is not cheaper than the current version of our method (with SWA-Gaussian) nor accurate. Just like the MC dropout, SWA-Gaussian does not incur any additional costs during training (except for a memory cost required for storing moving averages of parameters). More importantly, several works (Osband, 2016; Ashukha et al., 2020; Folgoc et al., 2021) pointed out that MC dropout underestimates the posterior uncertainty because it is approximating the posteriors with variational distributions of restricted form. The revised version of our paper now presents a more detailed algorithm in Appendix A.4. We believe that lines 1-10 of Algorithm 1 clarify the concerns about SWA-Gaussian.
>
> The most training complexity may come from the classifier learning stage (lines 11-17 of Algorithm 1) rather than the representation learning stage (lines 1-10 of Algorithm 1). Regarding this point, we already provided an analysis of the training costs of our method (requiring multiple forward passes during classifier learning) compared with vanilla decoupled training (requiring a single forward pass during classifier learning). To summarize, (i) the additional training cost is not that big, and (ii) it is worth spending such cost given the improvements.
>
>
> > The SOTA comparison in Table 4 seems to have unfair aspects: In some cases, less training epochs are used with the compared baseline (e.g. the comparison against KCL) and more are used in some others (Liu et al. (2021)).
>
> We fixed a typo in the comparison against KCL (i.e., the existing number stands for 200 epochs). Regarding the result of Liu et al. (2021), they virtually require more training costs since they applied Sharpness-Aware Minimization (SAM; Foret et al., 2021). SAM doubles forward and backward passes for each training iteration, and thus training ours with 400 epochs still requires less training costs (i.e., 400 < 600). Consequently, we believe that Table 4 can confirm the superiority of our method (i.e., ours achieves higher accuracy than baselines with fewer or almost equal costs).

---

> > ### Author Response · Authors · 2022-11-15
> > **Response to 1FXP (2/2)**
> >
> >
> > > The experimental evaluation is missing important comparisons with more recent approaches from 2022. No approach is compared against from 2022.
> >
> > As we replied in the general response, we further investigated whether our proposed method could combine with the existing state-of-the-art. Again, Appendix B.4 clearly shows the improvements after applying ours to the existing code bases.
> >
> >
> > > The experimental evaluation is missing comparisons on CIFAR10LT & CIFAR100LT. This is important to get a better feeling about how the method performs in a small-scale problem (CIFAR10LT) and a middle/large-scale problem such as CIFAR100LT. In fact, CIFAR100LT is surprisingly challenging for some problems and you can obtain unexpected results.
> >
> > We believe that our general response addresses this question. Also, a revised version of the paper mentions it in the main text so readers can easily find them (cf. Section 6.2). Thank you for your suggestion that can make our paper solid.
> >
> >
> > ----------
> >
> > References
> >
> > - (Mandt et al., 2017) Stochastic Gradient Descent as Approximate Bayesian Inference.
> > - (Maddox et al., 2019) A Simple Baseline for Bayesian Uncertainty in Deep Learning.
> > - (Gal and Ghahramani, 2016) Dropout as a Bayesian Approximation: Representing Model Uncertainty in Deep Learning.
> > - (Osband, 2016) Risk versus Uncertainty in Deep Learning: Bayes, Bootstrap and the Dangers of Dropout.
> > - (Ashukha et al., 2020) Pitfalls of In-Domain Uncertainty Estimation and Ensembling in Deep Learning.
> > - (Folgoc et al., 2021) Is MC Dropout Bayesian?
> > - (Liu et al., 2021) Self-supervised Learning is More Robust to Dataset Imbalance
> > - (Foret et al., 2021) Sharpness-Aware Minimization for Efficiently Improving Generalization

---

> > ### Comment · Reviewer_1FXP · 2022-11-16
> > **Thank you**
> >
> > Dear authors,
> >
> > Thank you for the detailed feedback and revision. I am quite happy with the revised version.
> >
> > > We would appreciate you introducing us to such papers you mentioned. It seems that the existing works (i) enhance the feature extractor by considering the under-represented classes during the representation learning stage, and thus, (ii) can obtain better decision boundaries.
> >
> > Sure, here are some papers that identify representations as the bottleneck:
> > "Distributional Robustness Loss for Long-tail Learning"
> > "ELM: Embedding and Logit Margins for Long-Tail Learning"
> > "Adjusting Decision Boundary for Class Imbalanced Learning"
> >
> > > If so, we believe that our method shares some spirits with them. Specifically, ours also (i) consider the under-represented instances having dispersed stochastic representations after the representation learning stage (with SWA-Gaussian), and thus, (ii) can obtain better decision boundaries by considering the under-represented instances during the classifier learning stage. As a result, both the existing works you mentioned and our work still serve the same purpose; obtaining better decision boundaries.
> >
> > I am not saying that they are not in the same spirit or representation learning and decision boundary are unrelated. I am trying to suggest that "the main bottleneck is the decision boundary" is providing a limited view of the problem. In fact, since the learned representations for the under-represented classes is sub-optimal, we try to compensate it by adjusting the decision boundary.
> >
> > Best

---

> > > ### Author Response · Authors · 2022-11-17
> > > **The 2nd response to 1FXP**
> > >
> > > We are pleased to resolve your main concerns via the first revision. Regarding the remaining issue, we now get the point of your comment, thanks to the list of papers you mentioned. Following your suggestion, we further revised Section 1; it now provides a wider perspective on the problem. Thank you again for your constructive comments.

---

> > > > ### Comment · Reviewer_1FXP · 2022-12-11
> > > > **Updated rating**
> > > >
> > > > Dear authors,
> > > >
> > > > Thank you for the revision and the additional experiments. It is good to see that your method can complement recent methods as well. I've raised my original recommendation.
> > > >
> > > > Best

---

> > > > > ### Author Response · Authors · 2022-12-12
> > > > > **Response to 1FXP**
> > > > >
> > > > > > Dear authors,
> > > > > > Thank you for the revision and the additional experiments. It is good to see that your method can complement recent methods as well. I've raised my original recommendation.
> > > > > > Best
> > > > >
> > > > > We are happy about your positive re-assessment of our work. Thanks again for all your constructive and dedicated comments for revision.

---

### Author Response · Authors · 2022-11-15
**General response**

We would like to thank you for the constructive feedback from all reviewers. In addition to answering each comment, we would like to address some main revisions made after the first rebuttal in this general response.

- __Appendix A.4__ now includes __the detailed algorithm of our proposed method__. We believe this would not only improve the presentation of the paper but clarify that ours does not require huge costs for training (along with __Appendix B.5__).
- __Appendix B.2__ provides __the results on CIFAR10/100-LT__, which are smaller than ImageNet-LT and iNaturalist2018. It clarifies that our proposed method performs well even for small-scale problems (thank you to Reviewer 1FXP for supplementing the parts we didn't think of).
- Since the proposed method is a simple classifier re-training method utilizing SWAG and self-distillation, we see no reason it would not work upon the existing state-of-the-art frameworks. Indeed, __Appendix B.4__ shows __the compatibility between ours and the state-of-the-art methods__.
- __Appendix B.5__ further provides insights on per-class weight norms and per-class marginal likelihoods after re-training the classifier as we proposed. It demonstrates that ours achieves both the uniform marginal likelihood and the well-calibrated prediction, even though it does not balance weight norms. This phenomenon suggests that __our proposed method works distinctly from the existing "balanced-norm" approaches__ for dealing with long-tailed data (as mentioned by Reviewer fqWB).
- __Appendix B.5__ further provides insights into dispersion in the context of long-tailed recognition. It demonstrates that the head class tends to have smaller dispersion and that __our method is thus suitable for long-tailed learning__.
- Instead of the existing "balancing strategies" paragraph, __Section 5__ now provides __the "knowledge distillation" paragraph__ as a related work. __Appendix A.1__ already covers the balancing strategies, and we believe it is worth clarifying how ours differ from existing studies related to knowledge distillation in the main text (thanks to Reviewer DCKB for mentioning the existing distillation-based method).

---

### Decision · Program_Chairs · 2023-01-20

**Decision:**

Accept: poster

**Justification For Why Not Higher Score:**

Concerns on minor accuracy improvement were the main blocker.

**Justification For Why Not Lower Score:**

I think the proposal **does** offer improvements in terms of calibration. Also, the idea to incorporate uncertainty estimation in the formulation for long-tailed learning is interesting and unusual (in a positive way).

**Metareview: Summary, Strengths And Weaknesses:**


The submission proposes an approach for long-tailed classification. Briefly, the proposed approach constructs a moving average of parameters from a periodic sample of parameter trajectory during training (SWA, Izmailov et al., 2018). The averaged parameters are used to construct the representation (backbone network). This is then followed by a classifier re-training step based on stochastic representation. The proposal is evaluated on standard long-tailed classification benchmark data (i.e., CIFAR10-LT, CIFAR100-LT, ImageNet-LT, iNaturalist-LT).

All reviewers agree that the submission is well written and has sufficient details for reproducibility.
Main comments from the four reviewers are

[+] Positive results (in terms of accuracy) over baselines (1FXP).

[+] Bringing in uncertainty estimation to the context of long-tailed learning is interesting (fqWB).

[-] While the authors added new results to Appendix B.4 in response to this comment, the added results did not sufficiently address the concern (DCKB). The concern was that improvements obtained from the proposed method are limited.

[-] Overall gains shown in experiments appear to be minor, especially when compared to the added computational burden (fqWB, DCKB, eyhi). While the authors have commented that there is no additional training cost in terms of compute, there is a concern on the increase in memory cost.

I tend to agree with clarification from the authors that the proposed approach only requires storing two additional copies of model parameters (for the first and the second moments), and thus the increase in memory cost should not be significant. Regarding minor improvements, firstly the reviewers mainly focused on accuracy gains, which were positive (albeit, small). However, more importantly, the proposed method shows good gains on ECE (expected calibration error), a point that seems to be missing in the discussion among reviewers. To my knowledge calibration is not a common aspect to consider under long-tailed learning settings because of the difficulty of the problem. Often, only balanced accuracy is of interest. Proposing a method that touches upon calibration may open up a new direction. The improvement on ECE also aligns with the proposed method which brings uncertainty estimation into the formulation. With these merits, I think the community will benefit from this work.

Recommendation: accept.


**Note From Pc:**

if the above contains the word "oral" or "spotlight" please see: "oral" presentation means -> notable-top-5% and "spotlight" means -> notable-top-25%. As stated in our emails, we are disassociating presentation type from AC recommendations

**Summary Of Ac-Reviewer Meeting:**

N/A